# The genetic framework of shoot regeneration in *Arabidopsis* comprises master regulators and conditional fine-tuning factors

Robin Lardon [1], Erik Wijnker[2], Joost Keurentjes [2] & Danny Geelen [1✉]

Clonal propagation and genetic engineering of plants requires regeneration, but many species are recalcitrant and there is large variability in explant responses. Here, we perform a genome-wide association study using 190 natural *Arabidopsis* accessions to dissect the genetics of shoot regeneration from root explants and several related in vitro traits. Strong variation is found in the recorded phenotypes and association mapping pinpoints a myriad of quantitative trait genes, including prior candidates and potential novel regeneration determinants. As most of these genes are trait- and protocol-specific, we propose a model wherein shoot regeneration is governed by many conditional fine-tuning factors and a few universal master regulators such as *WUSCHEL*, whose transcript levels correlate with natural variation in regenerated shoot numbers. Potentially novel genes in this last category are *AT3G09925*, *SUP*, *EDA40* and *DOF4.4*. We urge future research in the field to consider multiple conditions and genetic backgrounds.

[1] Department of Plants and Crops, Horticell Lab, Ghent University, 9000 Ghent, Belgium. [2] Department of Plant Sciences, Laboratory of Genetics, Wageningen University and Research, 6708 PB Wageningen, The Netherlands. ✉email: Danny.Geelen@UGent.be

Plant cells exhibit remarkable developmental plasticity, enabling them to reconstruct tissues and organs upon wounding during post-embryonic development[1,2]. This feature also allows for regeneration of entire plants from tissue explants cultured in vitro, which is widely exploited for clonal propagation of elite varieties, virus sanitization and the creation of transgenic crops[3,4]. As the success of such practices is highly variable among species and cultivars, many studies have focused on the model organism *Arabidopsis thaliana* to obtain insight into the molecular framework of de novo organ formation[3]. Here, shoots can be regenerated from root explants by a two-step protocol in which root segments are placed on auxin-rich callus-inducing medium (CIM), before being transferred to cytokinin (CK)-rich shoot induction medium (SIM)[5].

During CIM preincubation, auxin signals transmitted by the SCF–TIR1/AFB receptor complex, Aux/IAA repressors and auxin response factors (ARFs) activate division of designated pericycle cells to form a mass of organogenic callus[6,7]. This tissue resembles lateral root primordia and expresses root meristem genes such as *PLETHORA (PLT) 1 & 2*, *WUSCHEL-RELATED HOMEOBOX (WOX) 5*, *SHORT ROOT (SHR)*, and *SCARECROW (SCR)*[8,9]. The convergence of hormone signals (e.g., auxin-induced PLT3, 5 & 7/CUP-SHAPED COTYLEDON (CUC) 1 & 2 and WOX11/ LATERAL ORGAN BOUNDARIES DOMAIN (LBD) 16 modules[10,11]) with stress and wounding responses (e.g., mediated by WOUND-INDUCED DEDIFFERENTIATION (WIND) 1[12]) on CIM also underlies the acquisition of competence to regenerate shoots later on[13], by reactivating the cell cycle and installing progressive epigenetic changes such as DNA demethylation and histone modifications (e.g., H3K4me2 and H3K27me3)[14–17]. High cytokinin levels in the SIM then trigger a phosphorelay consisting of *Arabidopsis* histidine kinases (AHK2–4), phospho-transfer proteins (AHP1–5) and response regulators (ARRs) to repress root markers and induce coordinated expression of shoot apical meristem (SAM) genes such as *WUSCHEL (WUS)*, *SHOOT MERISTEMLESS (STM)*, *ENHANCED SHOOT REGENERATION (ESR) 1 & 2*, and *LIGHT-SENSITIVE HYPOCOTYLS (LSH) 3 & 4*[7,18]. This results in the "transdifferentiation" of root-like protuberances to shoot primordia, which is limited to a narrow developmental window[6,19,20].

The homeobox transcription factor (TF) WUS has been put forward as a master regulator of de novo SAM establishment[21]. Under steady-state meristem growth, it forms a feedback loop with *CLAVATA3 (CLV3)* to maintain the size and position of the stem cell niche in the SAM[22]. On SIM, *WUS* is induced following a two-step model that involves the dilution of repressive epigenetic marks by CK-induced cell division prior to direct activation by B-type ARRs[21,23–25]. The latter interact with HD-ZIP III TFs PHABULOSA (PHB), PHAVOLUTA (PHV), and REVOLUTA (REV) to spatially confine *WUS* expression and they limit auxin responses by blocking *YUC* transcription, further contributing to the elevated *WUS* levels[21,23]. WUS in turn reinforces CK responses by inhibition of A-type ARRs. PHB, PHV and REV also promote expression of the shoot determinants *STM* and *RAP2.6L* and WIND1 contributes to the events on SIM by directly activating *ESR1*[13,26].

Besides the type and physiological status of explants, incubation conditions such as hormone concentrations, temperature, and light affect regeneration and despite the conservation of cytokinin signals and *WUS* in the control of the SAM, natural *Arabidopsis* accessions show strong variation in their capacity to regenerate[2,27]. Previously, linkage mapping uncovered five quantitative trait loci (QTLs) responsible for the difference in regenerative capacity between accessions Nok-3 and Ga-0 and local association analyses resolved *RECEPTOR-LIKE PROTEIN KINASE (RPK) 1* as a potential candidate gene[28]. Here, we further

exploit natural variation in de novo shoot formation by conducting a genome-wide association study (GWAS), which is combined with the analysis of chromosome substitution lines (CSLs), T-DNA insertion mutants and gene expression levels. Known shoot meristem regulators and potential novel mediators of organogenesis are identified, confirming that regeneration is a complex trait controlled by multiple loci. We find that most rate-limiting factors are specific for the applied protocol and a few genes act as master regulators, including *WUS*, *AT3G09925*, *SUP*, *EDA40*, and *DOF4.4*.

## Results

**Shoot regeneration is subject to natural variation**. We have subjected 190 natural *Arabidopsis* accessions to the two-step protocol for shoot regeneration from root explants and scored various phenotypes that reflect the organogenic potential, including the number of shoots, shoot primordia and roots (details of the recorded traits are presented in the "Methods" section and Supplementary Fig. 1; Supplementary Fig. 2 provides corresponding bar charts). To incorporate the environmental effects, we have tested two protocol variants designated a and b, differing in explant age, light quality, and cytokinin concentration in the SIM. Irrespective of the protocol, the number of regenerated shoots follows an exponential trend, in which around half of the tested accessions do not form shoots at all (Fig. 1b). This similarity between the results of both protocols not only shows that the data are robust, but also indicates an important contribution by the genotype. A compact letter display based on nonparametric statistics further reveals that accessions can be divided over 14 and 19 different classes according to regenerated shoot numbers with protocol a and b, respectively. This multitude of levels in the phenotype can only be explained by numerous small allelic contributions, which suggests that de novo shoot organogenesis is a multigenic trait, a notion that agrees with the state of the art[13].

Plotting the maximum of regenerated shoot numbers under either protocol on a world map using accession coordinates reveals no obvious pattern (Fig. 1a), implying that if regeneration yields an evolutionary advantage, it is not coupled to geographic parameters. Besides, strong differences were recorded among explants of the same genotype (reflected by the error bars in Fig. 1b), indicating that there is an important residual effect likely contributed by the environment and the physiological state of explants. Hence, our data illustrate that higher order variation in de novo shoot organogenesis in *Arabidopsis* is explained by the genotype and variability within accessions is likely due to environmental fluctuations and epigenetic effects. Apart from variation in the extent of shoot regeneration, there is also a lot of diversity in the morphology of regenerated structures, going from hairy root-like outgrowths to trichome-covered shoots, leaf-like structures, and flower buds. Finally, the equal distribution of accessions analysed in different batches (represented by different colors in Fig. 1b) across the graph shows that there was no substantial blocking effect.

**Association patterns differ with environmental conditions**. To uncover the genetic framework of the observed regenerative variation, we computed trait correlations for single nucleotide polymorphisms (SNPs) across the genome (see "Methods" section), displayed in Manhattan plots (Fig. 2a, b and Supplementary Fig. 3). Multiple significant associations were found for regenerated shoot numbers (grouped in 5–10 distinct regions per chromosome), with an increased density of highly significant SNPs on chromosomes 3 and 2 for data from protocol a and b, respectively. Evaluating all genes within 10 kb of significant SNPs as potential quantitative trait genes (QTGs) pinpoints *URH1*

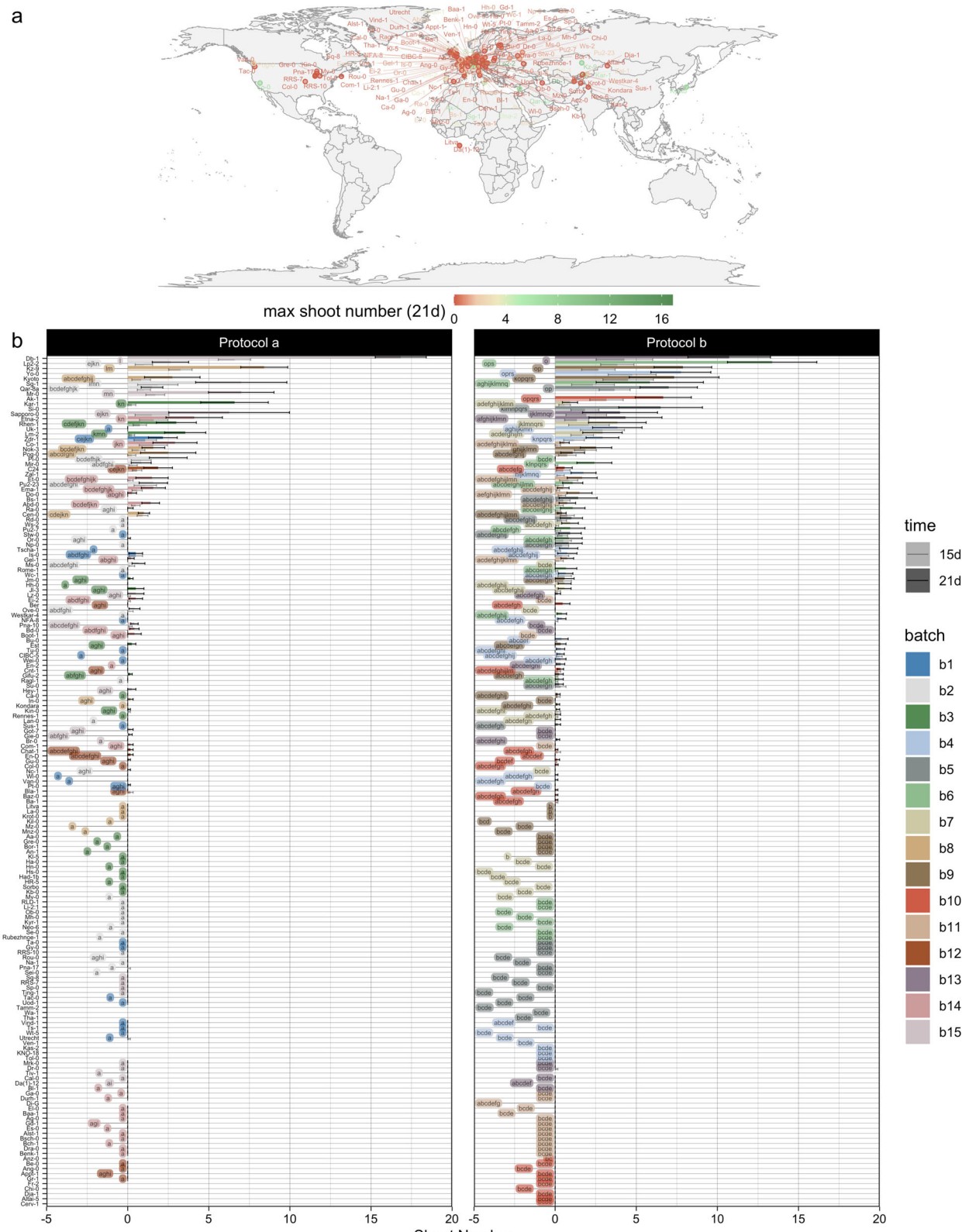

**Fig. 1 Variation in shoot regeneration among natural _Arabidopsis thaliana_ accessions. a** Geographic distribution of accessions ranked according to the maximum number of regenerated shoots after 21 days with protocol a or b. **b** Bar chart of shoot numbers after 15 and 21 days with protocol a and b. Colors indicate the batches in which accessions were analysed and labels reflect a compact letter display of two-sided pairwise comparisons (on count data after 21 days) using Dunn's nonparametric test at a false discovery rate (FDR) of 5% with $n = 12$ independent biological replicates. Global Kruskal–Wallis tests yielded $p$-values < 2e−16 for both protocols.

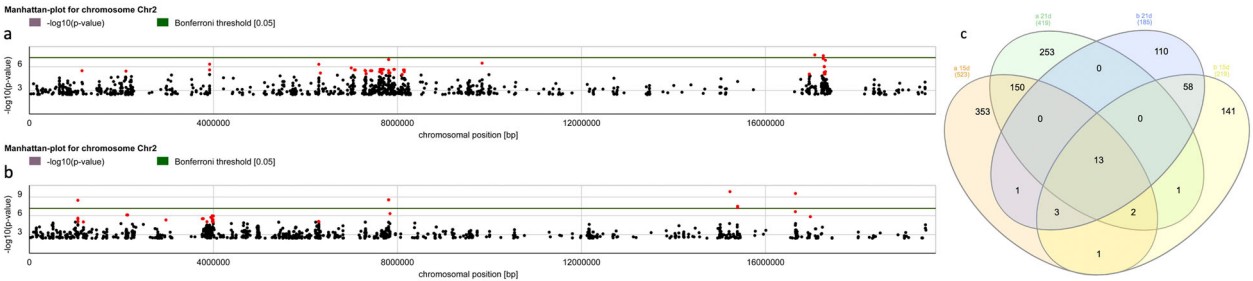

**Fig. 2 SNP associations for shoot regeneration. a, b** Manhattan plots of chromosome 2, showing the significance of SNP associations with regenerated shoot numbers after 21 days under protocol a (**a**) or b (**b**). The green line marks a Bonferroni threshold of 5%, deduced from an efficient mixed-model association expedited (EMMAX) test with $n = 129$ and $n = 149$ independent samples for protocol a and b, respectively. SNPs considered significant in subsequent analyses (raw p-value $\leq$ 1e−5) are highlighted in red. **c** Venn diagram showing overlap in genome-wide SNPs that were significantly linked to shoot numbers under protocol a (orange and green parts) or b (blue and yellow parts) after 15d (orange and yellow parts) or 21d (green and blue parts).

(or *SAP4*), *MIR393A* (or *MYB25*), *AT3G09925* (or *IPS1*), *SAUR23*, *WUS*, *QUL2* (or *EFM*), *ARF20*, and *MSL3* (or *RLP9*) as important players (Supplementary Data 1; refer to the "Discussion" for an in-depth review of candidate genes). This myriad of QTLs is in line with the genetic complexity expected from the phenotypic trend (Fig. 1b). Although there is partial overlap in association patterns under protocol a and b, the ranking of the SNPs is altered, suggesting that different factors are rate-limiting under different conditions. This is also observed in a Venn diagram of SNPs with $p \leq$ 1e−5 and MAF (minor allele frequency) > 5% for shoot numbers after 15 and 21 days following protocol a and b, showing that relatively few SNPs are highly significant for both procedures (Fig. 2c). The fraction of SNPs linked to regenerated shoots after 15 and 21 days under either protocol is higher, indicating a common regulatory network for de novo SAM establishment and subsequent shoot development.

Given the above, we explored the overlap in genes harboring allelic variation important for different phenotypes. Hereto, the genes closest to SNPs significantly associated ($p \leq$ 1e−5) with ten traits of interest were selected and a comparison of the subsets was made using the UpSetR package in R (Fig. 3a)[29]. Attribute plots were constructed to assess the significance, allele frequency, impact and number of SNPs underlying the genes in each category (Fig. 3b, c). Because few genes are found in higher-order intersections, Fig. 3a reveals that most candidates only play a role for specific traits under specific conditions. Moreover, intersections corresponding to different characteristics under the same protocol contain more genes than intersections spanning the two procedures, suggesting that critical determinants are shared between various regeneration pathways under similar conditions. However, candidates linked to root-like structures form a distinct category. Intriguingly, around ten factors are linked to many phenotypes across protocols and genes in highlighted intersections are supported by larger SNP clusters with low p-values (Fig. 3b). Moreover, their positive alleles are rare (low MAF) and often correspond to beta values at the edge of the distribution, meaning they contribute substantially to regenerative variation (Fig. 3c). The two genes found in most sets are *AT3G09925* (encoding a pollen ole e 1 and extensin family protein) and *IPS1*—likely corresponding to a single peak, followed by *WUS*, *SUP/FLO10*, *EDA40*, *DOF4.4*, *LCR85*, *AT2G13275*, *ACS10*, and *EMB2296*. Notably, many of these genes act in embryo or flower development (see "Discussion" section).

**GWAS highlights the importance of WUS and uncovers novel candidate genes.** Another strategy we followed to assign priorities to QTLs existed in ranking the allelic divergence of significant SNPs ($p \leq$ 1e−5 for shoot numbers after 21 days under protocol a or b) in the ten best and ten of the worst regenerating accessions

(Fig. 4). The most interesting candidates according to this criterium are *UBC28*, *QUL2*, *DREB1A*, *MIR393A*, *GWD2*, and *EDA40* (for which at least 6 out of 10 strong regenerators, but at most 1 out of 10 poor regenerators carries the beneficial allele). Moreover, of the 25 loci with the most pronounced allele distinction between good and bad performers, five are linked to at least four different phenotypes (*AT3G09925*, *WUS*, *EDA40*, *DOF4.4*, and *UXS5*; Supplementary Data 1). Looking at the allele distribution in Fig. 4 reveals that a variety of beneficial SNP combinations can lead to strong regeneration, whereas poorly regenerating accessions tend to have all negative alleles. In other words, the proposed candidates appear to provide more positive than negative selection, suggesting that they act as stimulators of regeneration rather than suppressors. Also, different sets of favorable SNPs are required for optimal performance under different protocols (i.e., there is a slight reorganization among top accessions under different conditions), implying a change in rate-limiting factors and epistasis effects depending on the environment.

Among the top candidates is *WUSCHEL* and a close-up of the association with shoot numbers under protocol b unveils three significant SNPs surrounding its open reading frame (ORF), 2 of which are located downstream and 1 is in the promoter sequence (67 bp from the transcription start site or TSS; Fig. 5a). They are highly significant, exhibit strong allelic distinction and exert a large phenotypic effect (Fig. 5a–c), independent of the conditions. As there are no nucleotide changes in the gene body and hence no potential effects on protein conformation, we investigated possible transcriptional regulation. RT-qPCR revealed three-fold higher mRNA levels in Lp2-2 than in Col-0 after 3 days on SIM (Fig. 5d) and as these are respectively good and bad regenerators, changes in *WUS* expression might indeed be responsible for natural variation in shoot regeneration. Nonetheless, our GWAS shows that various other factors contribute to the observed variability, which is likely a result of differential regulation at various stages of de novo shoot organogenesis, including founder cell specification, pluripotency acquisition, and SAM patterning.

**Chromosome substitution lines refine the GWAS.** Considering the genetic complexity of de novo shoot organogenesis and the possibility that epistasis contributes to variation between accessions, we phenotyped a full set of Col-0 × L*er* CSLs[30] using protocol b. Although the potential for mapping is restricted because of limited differences in the regenerative potential of Col-0 and L*er*, there is also limited genetic variation between them at the loci of interest and overlap with association data thus allows to refine certain QTLs. According to the results, lines carrying a L*er* chromosome 2 regenerate significantly better than the ones with a corresponding Col-0 variant (irrespective of the rest of the genome; Fig. 6a). Of the GWAS candidates found on this chromosome, only

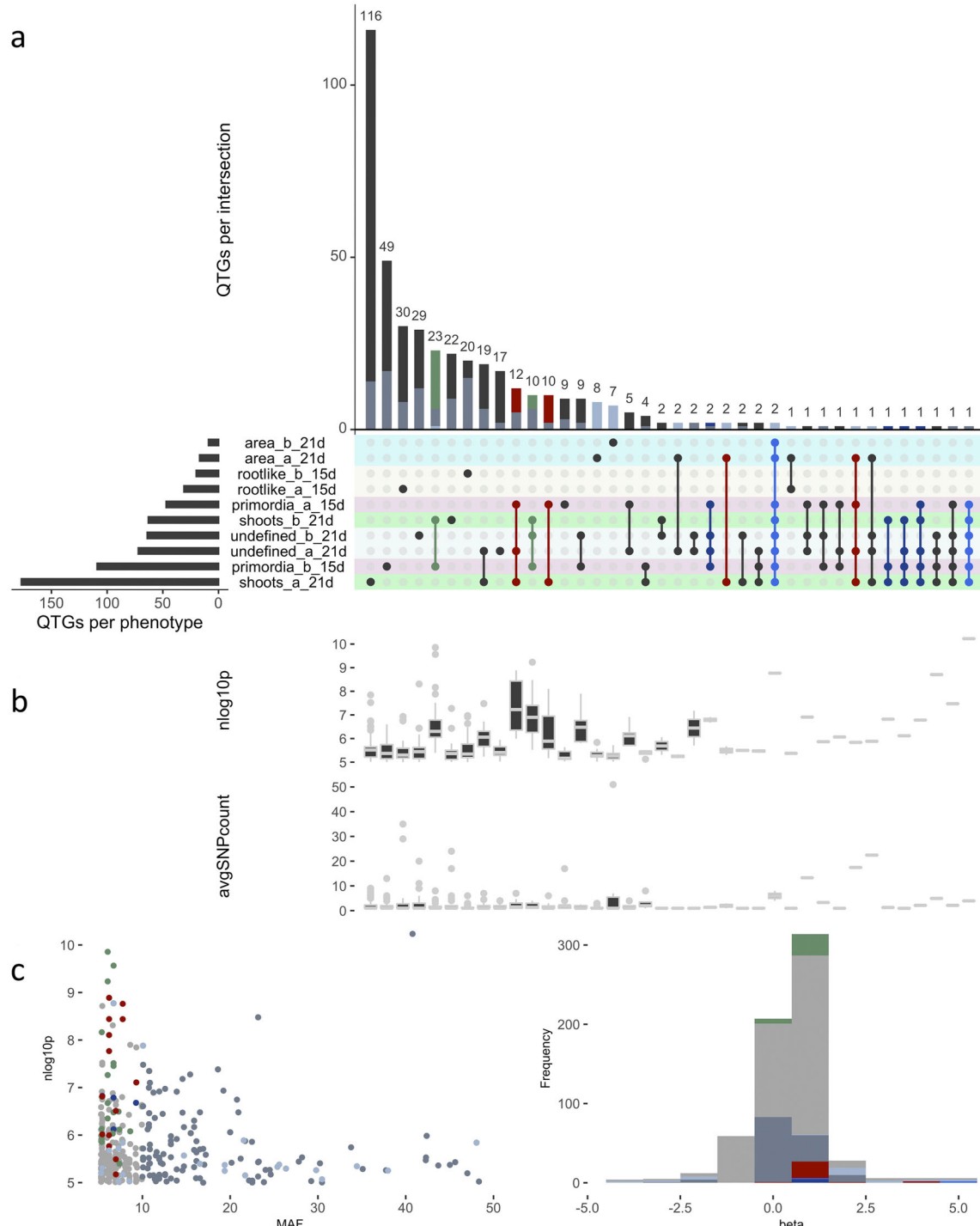

**Fig. 3 Overlap in candidate genes for various regeneration traits. a** Upset diagram showing overlap in QTGs (i.e., genes closest to SNPs with $p \leq 1e-5$) between phenotypes, protocols and time points. Bright blue, dark blue, red, and green colors respectively indicate the most common sets (degree $\geq 6$), selected intersections spanning both protocols and the highest order groups within protocol a or b. The same color codes are used for individual SNPs and genes, which are further highlighted in light or dark steel blue if beta (i.e., regression coefficient) $\geq 1.0$ or MAF $\geq 10\%$. **b** Box plots showing $-\log10(p\text{-value})$ (nlog10p) and number of SNPs (avgSNPcount) supporting the genes in each intersection (hinges reflect quartiles). **c** Scatter plot of SNP significance vs. minor allele frequency (MAF) and histogram of beta values.

*WUS* and *QUL2* carry different alleles in Col-0 and L*er*. As the positive allele of *QUL2* is found in the Col-0 variant, however, *WUS* is the most likely cause of variation between these accessions. Intriguingly, a nucleotide substitution at 341 bp upstream of the TSS (GA**G**T to GA**T**T) creates an additional ARR binding motif in the *WUS* promoter on L*er* chromosome 2, which agrees with the notion that transcriptional regulation of *WUSCHEL* could be underlying regenerative differences between accessions. Chromosome 1 appears to be important as well, because lines with a Col-0 variant form more shoot primordia than those with the L*er* version (Fig. 6a) and significant interactions with shoot regeneration were found between chr3:chr4 and chr3:chr5 (with respective FDRs of

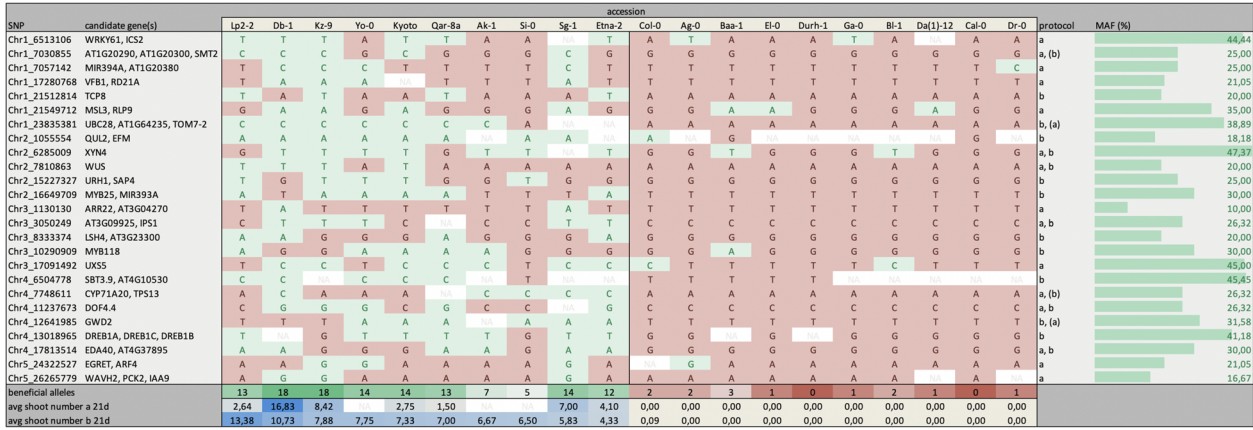

**Fig. 4 Allelic variation in the top candidate genes.** Distribution of selected SNPs among the ten best and ten of the worst regenerating accessions. Green and red boxes respectively indicate superior and inferior alleles and the three bottom rows depict gradients based on the accumulation of beneficial SNPs and regenerated shoot numbers after 21 days under protocol a or b. The two leftmost columns specify the SNP position along with nearby genes and the two rightmost columns show whether the SNP was significantly linked ($p \leq 1e-5$) to data from protocol a and/or b (with brackets reflecting insignificant associations) and the percentage of accessions carrying the minor and usually superior allele in the population (MAF). Blank fields contain missing data.

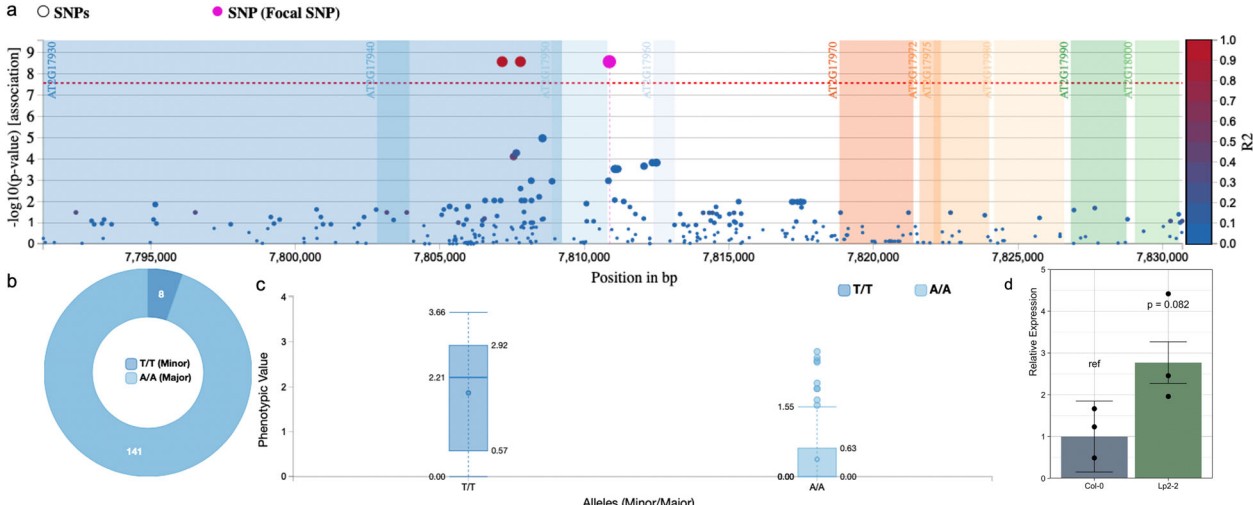

**Fig. 5 Details of the SNPs near *WUS* and their effect on shoot regeneration. a** SNP associations and linkage (R2) in the region around *WUS*, showing one upstream SNP (67 bp from the TSS; in pink) and two linked downstream SNPs (in red). **b** Allele frequencies of the pink SNP. **c** Box plot of the square root of regenerated shoot numbers after 21 days under protocol b in accessions with either variant of the highlighted SNP (hinges depict quartiles and whiskers extend to 1.5 times the interquartile range). **d** *WUS* expression relative to *UBC9* and *TIP41L* in Col-0 and Lp2-2 (respectively a poorly and a strongly regenerating accession) after 3 days on SIM following protocol b. Error bars reflect standard errors and the *p*-value (corresponding to an estimated effect size of −1.47 within a 95% confidence interval from −3.24 to 0.30) is deduced from a linear model using a two-sided ANOVA with n = 3 independent biological replicates (represented as black dots).

0.03 and 0.01, effect sizes of 42.60 and −61.82 and deviances of 6.2 and 11.1). Together with the small regenerative difference between Col-0 and L*er*, this suggests that *WUS* is not the only factor at play and variation between these accessions is orchestrated by a combination of positive and negative inputs. Ranking the CSLs reveals a complex pattern, wherein no particular interaction is highlighted (Fig. 6b).

**Novel candidate regeneration genes are highly context-dependent.** To validate the importance of novel candidates put forward by association mapping, we ordered T-DNA insertion lines for 25 genes, selected by significance, literature, specificity, allelic distribution and commercial availability (Supplementary Data 1). We obtained homozygous mutants for 11 of these genes (Supplementary Fig. 4) and analysed their phenotype using protocols a and b. In addition, a third protocol c was applied (similar

to protocol b, but with 6 days of CIM incubation and continuous light exposure during SIM) to improve comparison with wild type Col-0, which regenerates poorly using protocols a and b. Disruption of *AT3G09925*, *QKY*, *RLP9*, or *WAVH2* causes significant changes in de novo shoot formation (Fig. 7a), but none of the lines completely lost their regenerative capacity, potentially because of gene redundancy or incomplete loss-of-function. For comparison, we tested nine unrelated T-DNA mutants and five multiple gene knockouts linked to light signaling and phosphorylation, revealing that while the insertions in Fig. 7 are not as detrimental as higher order mutations, they yield more severe defects than random single gene disruptions (Supplementary Fig. 5). Visual inspection of the explants suggests that whereas *QKY* only affects callus growth, *AT3G09925* and *RLP9* are active at the stage of primordium formation and *WAVH2* determines both shoot initiation and development (Fig. 7b). However, *rlp9* shows decreased regeneration

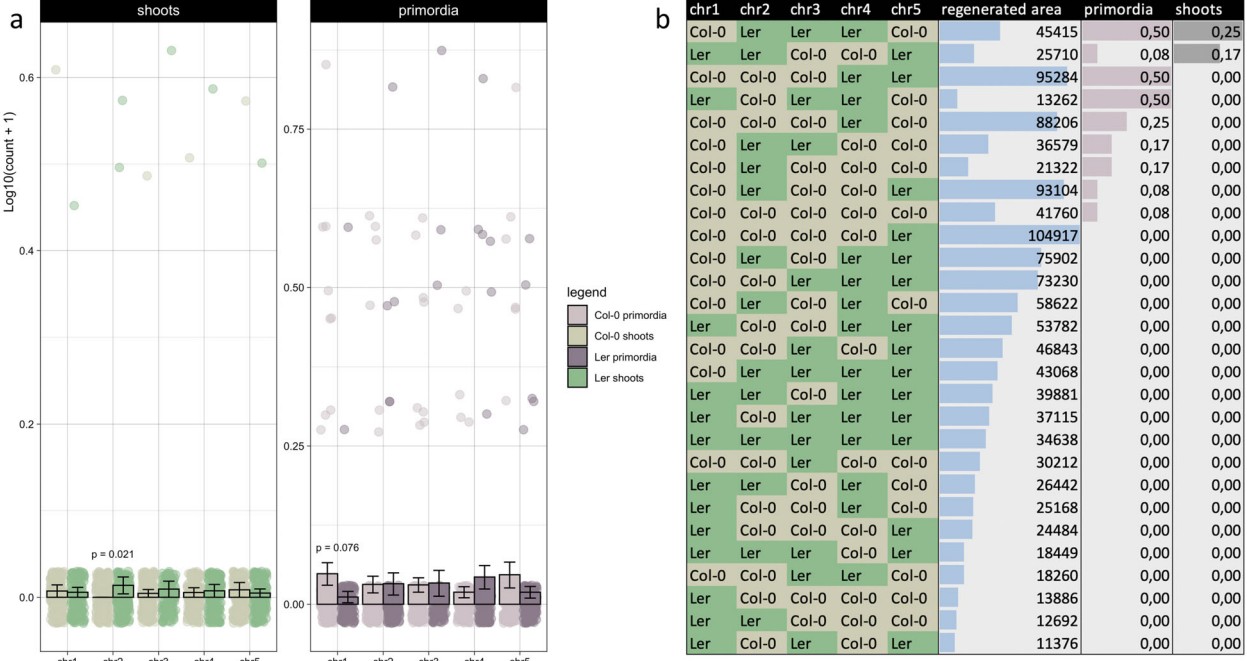

**Fig. 6 Shoot regeneration in chromosome substitution lines. a** Bar plot of log-transformed shoot and shoot primordium numbers in CSLs between Col-0 and Ler after 21 days on SIM following protocol b. Per chromosome, the averages of all lines with the Col-0 variant and the Ler variant are compared (irrespective of the other chromosomes) and individual data points are overlaid in jittered strip charts. Error bars reflect standard errors and FDR-adjusted $p$-values are deduced from negative binomial generalized linear models using two-sided likelihood ratio tests with $n = 12$ independent biological replicates per CSL and 321 or 318 residual degrees of freedom for shoots or primordia. The two significant terms had estimated sizes of 39.27 (effect of chr2 on shoots; deviance = 7.55) and −2.52 (effect of chr1 on primordia; deviance = 5.90). **b** Chromosome patterns of the CSLs (with Col-0 alleles in khaki and Ler variants in green), ranked by regenerated shoots, primordia and area.

under protocol a, while an improvement is detected under protocol b and the opposite holds for *at3g09925*. The effects of individual T-DNA insertions are also small compared to variation between protocols and although this could again be attributed to redundancy or weak null alleles, it suggests that single gene contributions are subordinate to environmental changes and that a combination of multiple alleles accounts for differential regeneration among accessions. Notably, protocol c yields much better regeneration rates in different wild type backgrounds and in many mutants. These results illustrate that many determinants of regeneration are highly context-dependent and thus we propose that only a small set of master regulators such as *WUS* are critical under various conditions.

## Discussion

In line with previous reports, our extended and genome-wide association analysis found substantial variation in the regenerative potential of 190 *Arabidopsis thaliana* accessions under two different protocols[28,31]. This agrees with the finding that root hormone levels differ among accessions[32], because hormone responsivity is a key determinant of regeneration[2,33]. *In planta* developmental traits such as rosette morphology, leaf expansion, and flowering time are subject to natural variation as well[34–36], implying that care must be taken when extrapolating observations regarding plant development made in a particular ecotype, but also highlighting the potential of GWAS. Association mapping revealed de novo shoot organogenesis to be a complex trait, similar to what was reported for in vivo shoot development[34]. It is controlled by several QTGs, including ARFs and ARRs, MYB and AP2/ERF2 family TFs, miRNAs, receptor-like kinases, F-box proteins, chromatin remodelers and various biosynthetic and cell wall modifying enzymes. Based on literature, we recognized 18 a priori candidates, 17 homologs of prior candidates, 19 genes involved in similar processes, and 75 unrelated or unknown genes. Plotting the number of phenotypes these factors support against a score for prior links with organogenesis revealed four categories (Fig. 8), which are elaborated below. Combined with the analysis of T-DNA mutants, Fig. 8 reveals a major group of fine-tuning factors (~95%) whose role depends on the protocol and a minor group of master regulators (~5%), that are critical for multiple tested procedures and traits. This shift in rate-limiting factors depending on the conditions could explain why several studies attempting to map natural variation in shoot regeneration have pinpointed different QTGs[28,31]. Accordingly, a GWAS on embryonic callus regeneration in maize showed that only 15 out of 63 QTNs were retained in multiple environments and highlighted *WUSCHEL-RELATED HOMEOBOX 2 (WOX2)*, although other candidates are distinct from ours[37]. Most QTGs we identified are also new compared to association studies on adventitious shoot regeneration in roses[38], callus formation in poplar[39] and rice[40] and in vitro regeneration of cucumber[41] and tomato[42]. However, these studies do report similar functional classes of candidates (e.g., embryogenesis and meristem genes, reprogramming factors, hormone-related proteins, receptor-like kinases, and TFs from the LBD, ERF, MYB, and WOX families)[37–40] and in cases where multiple traits, protocols or techniques are evaluated, overlap between them is limited[37,38], suggesting that the difference in experimental systems could be part of the cause. Notably, several established SAM genes, epigenetic factors and cell cycle regulators (e.g., STM, CUCs, ESRs, PLTs, WIND1, MET1, and CYCD3[13]) were not detected in our assay, which might be due to a lack of functional sequence variation at these loci in the tested population[43]. In turn, this could be the result of stringent selection against harmful mutations in genes that are vital to embryonic development, wound repair, and rooting.

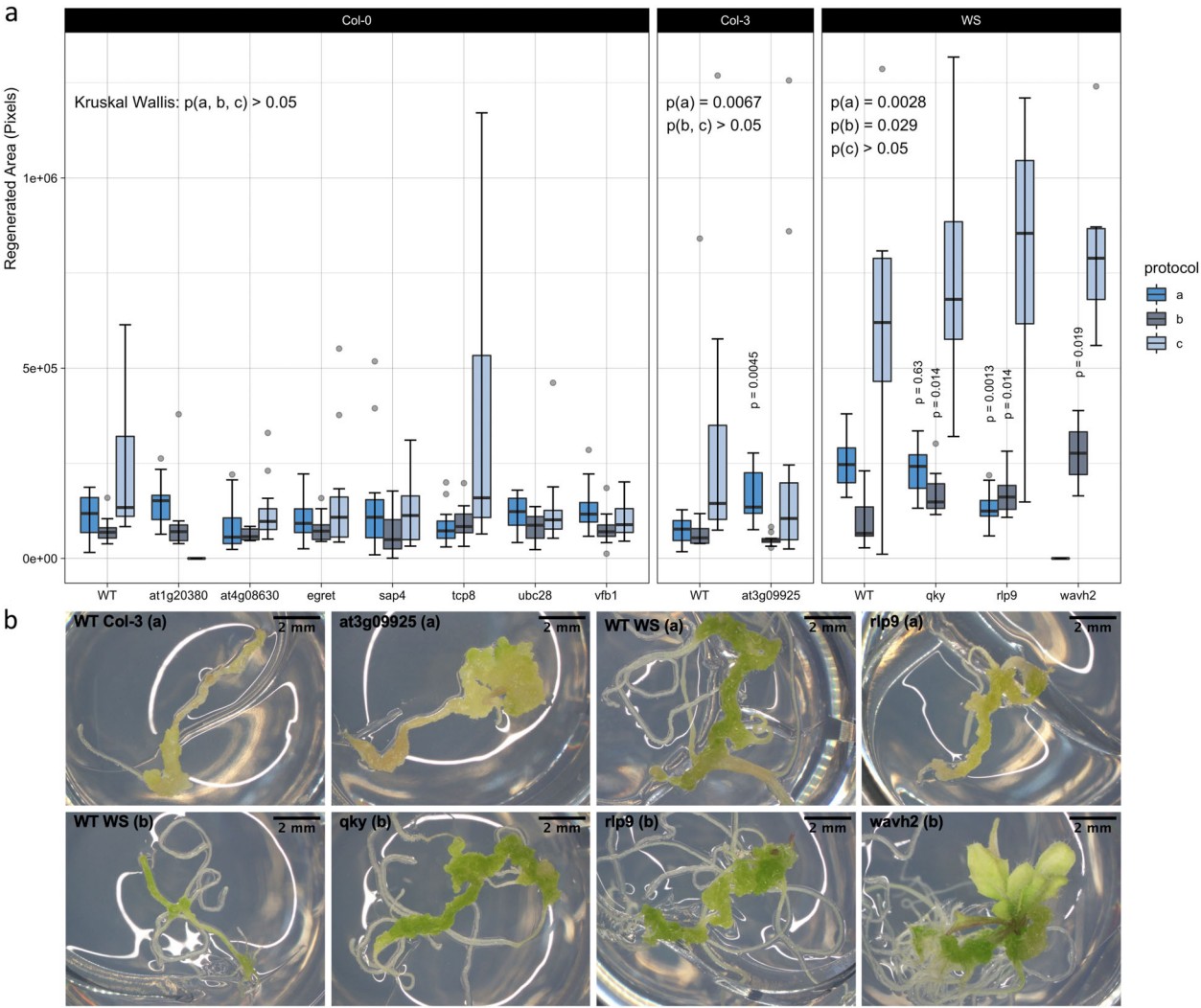

**Fig. 7 Shoot regeneration in mutants for GWAS candidate genes. a** Box plot of the regenerated area in 11 T-DNA insertion lines compared to wild type (WT) plants after 21 days under protocol a, b, or c (respectively shown in bright blue, dark steel blue, and light steel blue). Hinges mark the first and third quartile, horizontal lines reflect the median and whiskers extend to the furthest data point within 1.5 times the interquartile range of the nearest hinge. FDR-adjusted *p*-values are deduced from a Kruskal–Wallis test and Dunn's many-to-one tests for two-sided pairwise comparisons to a single control on $n = 12$ independent biological replicates (quantile estimates are 3.18, −0.47, 2.59, −3.60, 2.59, and 2.42 for significant terms from left to right, i.e., *at3g09925* (a), *qky* (a, b), *rlp9* (a, b), and *wavh2* (b)). No data was available for *at1g20380* under protocol c and *wavh2* under protocol a. **b** Representative images for lines that differ significantly from their wild type counterparts under the respective conditions (protocol a, b or c).

Possibly, epigenetic, transcriptional or post-translational regulation is favored for key survival genes to allow for better fine-tuning. Investigating the role of these mechanisms in natural regenerative variability by means of eQTL mapping and methylome-wide associations is a promising future prospect[44].

A strong correlation was found between regenerated shoot numbers and allelic variation in the promoter of *WUSCHEL*, and we propose that this is due to differential transcription. *WUS* is essential for formation and maintenance of the SAM and knock-out of this gene severely impairs regeneration[18,22]. Moreover, its expression marks shoot progenitor cells on SIM and overexpression induces somatic embryogenesis and shoot regeneration[21,45]. Hence, this association demonstrates the robustness of the GWAS. CSL analyses suggest that *WUS* could also contribute to regenerative differences between Col-0 and L*er*, and sequence comparison uncovered a SNP that introduces an additional ARR binding motif in the promoter of the beneficial L*er* allele (GA**G**T to GA**T**T; 341 bp upstream of the TSS). This SNP is also present in Lp2-2, a strong regenerator with 3-fold higher *WUS* mRNA levels on SIM than

poorly regenerating Col-0 plants, which is in line with recent reports showing direct *WUS* induction by B-type ARRs[21,23]. However, at present we cannot distinguish whether improved shoot formation in Lp2-2 is due to an increased number of WUS-expressing foci or elevated WUS levels in individual foci. Lastly, the SNP upstream of *WUS* overlaps with one of four unresolved QTLs obtained by linkage mapping[28].

Two other prime candidates are *LSH4* and *CLE2*, whose linked SNPs show high trait specificity and are downstream of the ORF. *LSH4* is induced by *CUC1* in the boundary cells of shoot organs to coordinate differentiation and overexpression leads to ectopic development of *WUS*-expressing meristems[46]. It has also been used successfully as a marker for shoot regeneration[47]. *CLE2* is a homolog of CLV3, known for its role in the regulatory feedback loop with WUS that controls the size of the SAM *in vivo*[48]. It is induced by *ESR1* during the early stages of regeneration[49] and upregulated in *ick1/2/5/6/7* mutants showing increased regenerative potential[50]. E2FB, another TF we picked up, is also regulated by *ICK/KRP* genes[51] and acts in the translation of environmental

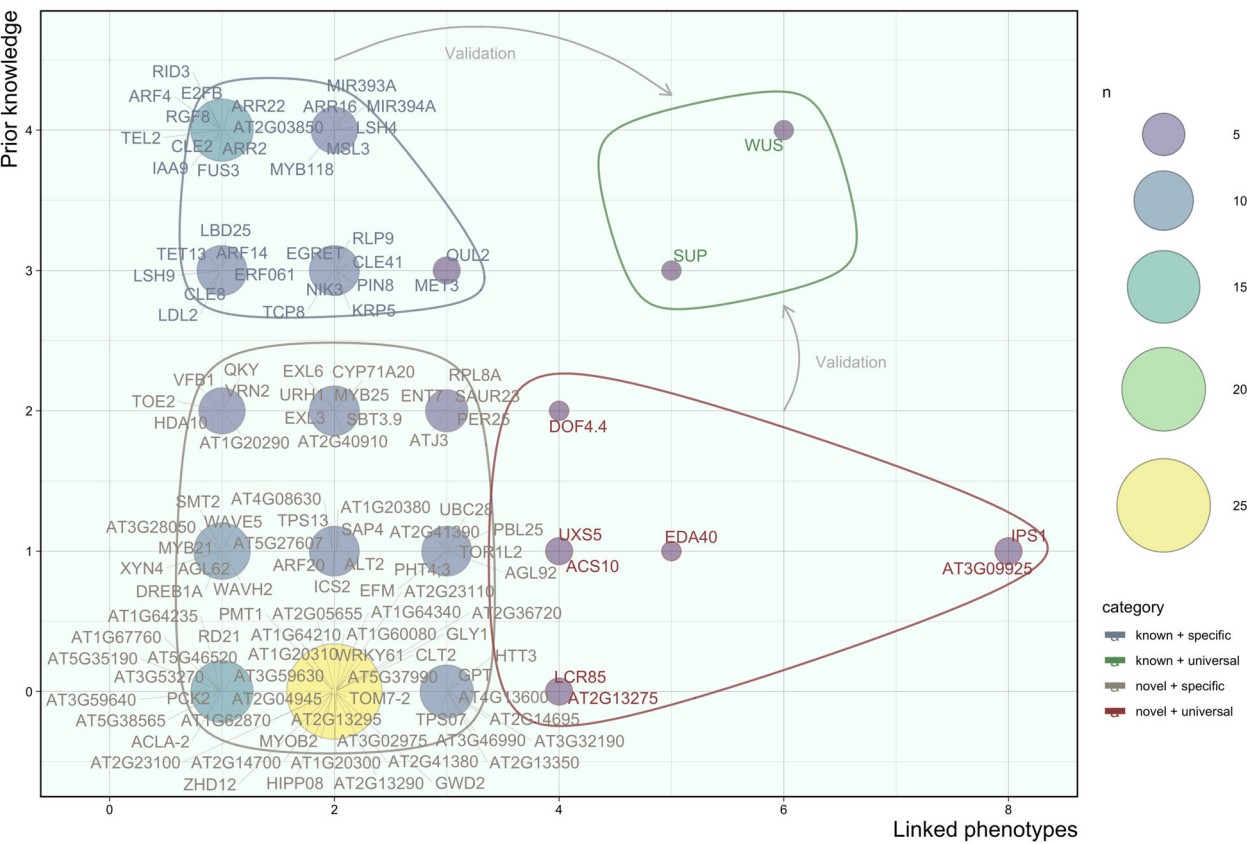

**Fig. 8 Visual representation of the four categories of candidate regeneration genes.** The number of associated phenotypes, i.e., shoots, shoot primordia, green area, root-like structures, and undefined structures under protocol a or b (x-axis) was plotted against prior links with shoot regeneration based on literature, where 0 = no link, 1 = possible link, 2 = plausible link, 3 = indirect link, 4 = direct link (y-axis). The size and color of circles reflects the number of genes in each set (n). Blue, green, gray and red outlines respectively define known conditional factors, known master regulators, unknown conditional factors, and unknown master regulators.

stimuli and auxin signals to cell cycle progression in root and shoot apexes[52,53]. In addition, we found associations near two micro-RNAs: MIR393A (overexpression of which impedes de novo SAM formation by repressing TIR1[54]) and MIR394A (known to spread from the L1 layer of the SAM to the L3 layer, where it represses LCR to potentiate WUS activity and therefore maintain stem cell pluripotency[55]). Both miRNAs are differentially expressed between totipotent and non-totipotent calli[56]. We also found two auxin-related and three cytokinin-related prior candidates: IAA9 is upregulated during CIM[57] preincubation and implicated in lateral root formation[58] and somatic embryogenesis[59], while ARF4 acts redundantly with ARF3 to control organ polarity[60] and lateral root initiation[61]. Recently, it was discovered that these ARFs undergo differential methylation by MET1 during shoot regeneration[62], and they promote organogenesis by repressing STM through histone deacetylation[63]. On the other hand, ARR2 is reported to associate with HD-ZIP III TFs to control WUS expression during regeneration and impair the process when knocked-out[21], whereas ARR16 and ARR22 are A-type response regulators. The former is downregulated during CIM preincubation and induced by ARR2 on SIM[9,64], and the latter suppresses B-type ARRs[65]. Seven lesser-known prior candidates mainly involved in lateral rooting, callus formation or somatic embryogenesis are MSL3[66], MYB118[67], RGF8[68], RID3[69], FUS3[70], TEL2[71], and AT2G03850[9].

Among indirect QTGs (i.e., homologs of prior candidates) is the pectin methyltransferase QUL2/PMT5, a paralog of QUA2/TSD2 (expressed in meristems and mutation of which causes

"shooty" callus formation in vitro, paralleled by enlarged expression domains of KNAT1&2 and elevated transcription of STM[72]). It is near six highly significant SNPs, but corresponding T-DNA lines were not viable. Another indirect candidate, TCP8, harbors a missense variant with large phenotypic effect and low p-value. Although TCP8 (a class I TCP) has only been linked to immunity, auxin homeostasis and leaf development[73–75], repression of TCP3 (a class II TCP) yields ectopic shoot formation, because it suppresses CUC genes and KNOX1 factors through interaction with AS2[76–78]. Moreover, strong redundancy is observed within and between TCP clades and class I TCPs can also modulate KNOX1 and cell cycle gene expression to control lateral organ growth. Some even affect cytokinin sensitivity or bind to AHPs[75,77,79]. Accordingly, a T-DNA mutant showed increased regeneration with protocol c. Next, ERF061 belongs to the DREB A-6 subfamily of ERF/AP2 TFs also containing WIND1 (able to bypass wounding and auxin pre-incubation and enhance callus formation and shoot regeneration by upregulating ESR1)[26,80]. This functionality is conserved in WIND2-4[80], but ERF061 is only supported by one downstream SNP. Two further indirect QTGs compete with a direct candidate: RLP9 is a homolog of CLV2/RLP10 (promoting stem cell differentiation by repressing WUS in intact SAMs[81]) that lies upstream of six SNPs in the gene body of MSL3 (mutation of which induces callus production at the shoot apex, coupled to an altered cytokinin to auxin ratio[66]). Nonetheless, disruption of RLP9, respectively decreased and increased the number of regenerated shoots under

protocol a and b, so both genes might contribute to the association. Likewise, EGRET/IDD13 is a C2HC zinc finger protein from the same family as JKD and MGP (known for their roles in root patterning[82]), but regeneration was not affected in egret mutants, suggesting that nearby ARF4 is causing the correlation. Lastly, LDL2 is homologous to LDL3, recently found to eliminate H3K4me2 during callus formation and thus facilitate the induction of shoot markers on SIM[17].

Some candidates are plausible QTGs because they act in processes related to shoot regeneration. For example, QKY is a component of SUB signaling required for tissue morphogenesis and organ development, mutation of which alters the morphology of the L2 SAM layer[83,84] and T-DNA insertion significantly increased regeneration under protocol b. Another such QTG is VFB1, encoding an F-box protein from the same family as TIR1 and AFB1-5. Loss of all four VFB genes compromises lateral root formation and DR5:GUS expression, but VFB2 cannot functionally substitute TIR1[85], which agrees with our finding that disruption of VFB1 had no significant effect. SUP/FLO10 (a TF that defines the boundary between stamens and carpels in the floral meristem by regulating cell proliferation and floral homeotic genes such as AP3 and PI[86]), DOF4.4 (related to shoot branching[87]) and EMB2296 are plausible QTGs that underlie at least three investigated phenotypes. Other candidates in this category worth mentioning are URH1 (potentially involved in CK metabolism[88]), ENT7 (a possible CK transporter[89]), AT1G20290 (a SWI-SNF-related chromatin-binding protein), VRN2 (a Polycomb protein), and HDA10 (a histone deacetylase).

Finally, a possible link was found for a few poorly annotated candidates: AT3G09925 is a Pollen Ole e 1 (POE1) allergen and extensin family gene and recent reports suggest that these genes could act in various aspects of plant development, as they exhibit specific transcription patterns and some members are even regulated by H3K27me3 (which also restricts WUS during SIM incubation)[16,90]. According to our data, this gene is the second most important regulator of regeneration (after WUS), because it harbors nine highly significant SNPs (in the ORF and downstream region) that contribute strongly to variation in eight out of ten recorded traits and show pronounced allelic differences between poor and strong regenerators. Curiously, T-DNA insertion had a slightly negative effect under protocols b and c, but it increased the regenerated area using protocol a. Another less obvious candidate is WAVH2, T-DNA disruption of which significantly increased regeneration under protocol b. Along with homologs WAV3 and WAVH1, it has been attributed a role in root gravitropism and phototropism (triple wav3 wavh1 wavh2 mutants showing abnormal auxin signals in the root)[91]. Intriguingly, another member of this gene family named EDA40 underlies five of the recorded phenotypes in our GWAS and shows strong allelic distinction between good and bad accessions. Hence, the WAVY GROWTH E3 ligases likely play a role in de novo organogenesis. The E2 ubiquitin-conjugating enzyme UBC28 is interesting because transcriptomic comparison of accessions with the beneficial allele against those with the weak variant revealed downregulation of this gene in the good accessions (Supplementary Data 2), but no significant changes were found in the T-DNA line. Other possible candidates are DREB1A/CBF3 (involved in cold stress[92]) and ACS10 (an aminotransferase without ACC synthase activity[93]).

Our take-home message is that natural variation in tissue regeneration is associated with allelic differences in master meristem regulators and conserved genes that play a role in embryogenesis and flower development, but only a minority of these genes have broad functionality. Most regeneration determinants are context-dependent, which also holds for novel candidates put forward by GWAS, whose precise role in de novo SAM formation remains to be elucidated. Because in vitro responses are highly variable among accessions and rate-limiting molecular factors depend on the applied protocol, we urge future research in the field of regeneration to consider multiple conditions and validate results in different genetic backgrounds.

## Methods

**Plant materials and growth conditions.** The GWAS included 190 natural Arabidopsis thaliana (L.) Heynh. accessions sequenced by the 1001 Genomes Consortium[94] (N76636; NASC). Col-0 × Ler chromosome substitution lines were kindly provided by Cris L. Wijnen, Erik Wijnker, and Joost Keurentjes (Wageningen University)[30]. T-DNA insertion mutants were retrieved from NASC (SALK and SAIL lines) and INRA (FLAG lines): SALK_121407C (N654861; at1g20380), SALK_012673C (N670321; at4g08630), SALK_044769C (N674790; egret), SALK_152010C (N681875; sap4), SAIL_656_F11 (N862668; tcp8), SALK_040325C (N653158; ubc28), SALK_128933C (N667298; vfb1), SAIL_208_E09C1 (N867008; at3g09925), FLAG_124D07 (DSH18; qky), FLAG_056E09 (DLL1; rlp9), FLAG_109A12 (DYB211; wavh2). T-DNA inserts were checked by PCR (Supplementary Fig. 4) using the primers in Supplementary Table 1.

Seeds were sterilized by exposure to chlorine gas for 4 h and sown on Gamborg B5 medium (3.1 g B5 salts including vitamins per liter of medium, with 0.05% 2-(4-morpholino)-ethane sulfonic acid (MES), 2% (w/v) glucose and 0.7% agar at pH 5.8) and vernalized for 4 days at 5 °C. Three procedures were used for shoot regeneration, respectively designated as protocol a, b and c (adapted from the work of Valvekens et al.[5]). In protocol a, seedlings were grown under cool white fluorescent tungsten tubes (70 μM m$^{-2}$ s$^{-1}$) at 21 °C following a 14/10 h light/dark regime. After 7 days, 7 mm long root segments including the tip were excised and placed on CIM (B5 supplemented with 2.2 μM 2,4-dichlorophenoxy acetic acid (2,4-D) and 0.2 μM kinetin) for 4 days. Next, the explants were transferred to SIM (B5 supplemented with 25 μM 2-isopentenyl adenine (2-IP) and 0.86 μM 3-indole acetic acid (IAA)) and incubated for 3 weeks. Protocol b differs from protocol a in three ways: warm white light was used (70 μM m$^{-2}$ s$^{-1}$), explants were excised after 10 days and only 5 μM 2-IP was used in the SIM. Protocol c is a slight modification of protocol b, whereby explants were kept on CIM for 6 days and they were placed in continuous light during SIM incubation. Pictures were taken under binoculars (6.3×) after 15 days and after 21 days.

**Genome-wide association study.** Accessions were randomly divided over 15 batches (15–40 accessions per batch) and within batches the regenerative capacity was assessed following either protocol a or b, in such a way that all accessions were subjected to both protocols. Pictures were taken from every well after 15 days and 21 days and used to manually count shoots, primordia, root-like structures and undefined structures (representative images showing how these features were distinguished are provided in Supplementary Fig. 1) and assign a score from 0 to 5 for callus formation and greening. Raw data were processed in R (version 3.5.2; Supplementary Code) and multiple secondary phenotypes were calculated, including the regeneration index (defined as the percentage of explants forming at least 1 shoot), shoot variability (measured as the standard deviation of shoot numbers divided by the regeneration index) and shoot formation rate (the number of shoots formed per day between 15 days and 21 days of SIM incubation). Images were also analysed with ImageJ (version 2.0.0-rc-69/1.52j; Supplementary Code) to determine the area of green structures as a proxy for the regenerative potential. Finally, trait averages per accession were calculated and data were curated to handle missing sequence data, incomplete germination or contamination (requiring at least six explants per accession) and a distributional error at the stock center[95], leaving 129 and 149 observations for the association analysis with protocol a and b, respectively (for Zal-1, Tol-0, KNO-18, Kas-2, Ven-1, Yo-0, Bu-0, Bs-1, Wa-1, Tamm-2, Su-0, Si-0, Tha-1, Ak-1, Cerv-1, Ba-1, Altai-5, Dja-1, Baz-0, Chi-0, Fr-2, Anz-0, and Di-G, no data was available under protocol a and for Tiv-1, Hey-1 and Mr-0, no data was available under protocol b). Note that for each phenotype, accessions showing excessive variability were also eliminated. Corresponding bar plots are provided in Supplementary Fig. 2.

**Bioinformatics.** For every phenotype described above, accession averages were uploaded to easyGWAS for correlation analyses[96]. A square root transformation was applied in the case of count values to approximate a normal distribution and three principle components were included in the model to account for higher order population effects. The minor allele frequency (MAF) was cut off at 5% and the EMMAX algorithm was used for computation. Resulting Manhattan plots and a list of candidate genes selected using custom python code (version 2.7.15; Supplementary Code) can be found in Supplementary Fig. 3 and Supplementary Data 1, respectively. Individual transcriptome analyses were performed in R for each of 297 manually selected candidate SNPs by considering accessions with identical alleles in that position as biological replicates and contrasting the two variant groups using limma (after converting counts to FPKM values)[97] to get differentially expressed genes (Supplementary Data 2; Supplementary Code). Three principle

components were incorporated in the design matrix to correct for population structure.

**RT-qPCR.** Three biological replicates of *Arabidopsis thaliana* Col-0 and Lp2-2 root segments were sampled after 3 days on SIM following protocol b (~100 mg fresh weight/sample). RNA extraction was done with the Qiagen RNeasy Plant Mini Kit and cDNA synthesis was done with the Promega GoScript$^{TM}$ system using random hexamer primers. Yield and purity were assessed by nanodrop. *WUS* expression was normalized to *UBC9* and *TIP41L* levels and a sample maximization strategy was applied for the plate layout (assays were spread over three runs, including two technical replicates, no-RT controls, and triplicate no-template controls). RT-qPCR was set-up with the GoTaq® master mix using primers described in Supplementary Table 1 and run on a Stratagene Mx3005P cycler. Results were analysed according to the ΔΔCt method using the R package pcr[98] (Supplementary Code).

**Statistics and reproducibility.** For phenotyping, 12 explants were analysed per combination of line (i.e., accession, CSL, or T-DNA insertion mutant) and protocol, divided over two sets of six that were incubated on multi-well plates according to a completely randomized block design. This number was determined by prior experience, feasibility and the use of 24-well plates (with two lines per plate). Several accessions were phenotyped repeatedly to confirm that objective phenotypic values were obtained. Association mapping was performed using a mixed-linear model on square-root-transformed phenotypes to correct for population stratification and data distribution. Count and area data were analysed using two-sided likelihood ratio tests on negative binomial generalized linear models or consecutive global and pairwise nonparametric tests (i.e., two-sided Kruskal–Wallis and Dunn's tests). Triplicate RT-qPCR data were analysed according to the ΔΔCt method and ANOVA on a linear model. Details of the statistics for each figure are provided in the corresponding caption.

**Reporting summary.** Further information on research design is available in the Nature Research Reporting Summary linked to this article.

## Data availability

The data generated and/or analysed in the current study are either included in this article as Supplementary Data or submitted to public repositories. Raw phenotype data (Fig. 1 and Supplementary Fig. 2) are available from AraPheno at https://doi.org/10.21958/study:80[99] and full GWAS data (Figs. 2, 5a–c and Supplementary Fig. 3) are available in easyGWAS at https://easygwas.ethz.ch/gwas/myhistory/public/24/ (accession codes for phenotypic averages underlying these results are AT1P23868, AT1P24025, AT1P24028, AT1P24031, AT1P24043, AT1P24048, AT1P24065, AT1P24068, AT1P24071, AT1P24088, AT1P25078, AT1P26148). Associations for raw phenotypes submitted to AraPheno were also recomputed using a permutation-based pipeline and published in the AraGWAS Catalog (https://doi.org/10.21958/gwas:1290, https://doi.org/10.21958/gwas:1283, https://doi.org/10.21958/gwas:1276, https://doi.org/10.21958/gwas:1269, https://doi.org/10.21958/gwas:1288, https://doi.org/10.21958/gwas:1281, https://doi.org/10.21958/gwas:1274, https://doi.org/10.21958/gwas:1267, https://doi.org/10.21958/gwas:1289, https://doi.org/10.21958/gwas:1282, https://doi.org/10.21958/gwas:1275, https://doi.org/10.21958/gwas:1268, https://doi.org/10.21958/gwas:1291, https://doi.org/10.21958/gwas:1284, https://doi.org/10.21958/gwas:1277, https://doi.org/10.21958/gwas:1270, https://doi.org/10.21958/gwas:1294, https://doi.org/10.21958/gwas:1287, https://doi.org/10.21958/gwas:1280, https://doi.org/10.21958/gwas:1273, https://doi.org/10.21958/gwas:1292, https://doi.org/10.21958/gwas:1285, https://doi.org/10.21958/gwas:1278, https://doi.org/10.21958/gwas:1271, https://doi.org/10.21958/gwas:1293, https://doi.org/10.21958/gwas:1286, https://doi.org/10.21958/gwas:1279, https://doi.org/10.21958/gwas:1272). For questions on data availability, contact robin.lardon@ugent.be.

## Code availability

Custom code for image processing (ImageJ), handling phenotypic data (R), selection of candidate genes based on association data (python), transcriptome and RT-qPCR analysis (R) is available in the Supplementary Code (plain text).

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

## Acknowledgements

We thank Hoang Khai Trinh for helping with experiments requested by the referees, Tim De Meyer and Wim Van Criekinge for advice on the computational analyses and Monica Höfte for sharing equipment. This work was supported by the Research Foundation Flanders (Fonds Wetenschappelijk Onderzoek; FWO), project numbers 1S48517N and G094619N.

## Author contributions

R.L. designed and performed experiments, analysed and visualized data and wrote the manuscript. D.G. conceptualized research, supervised the project and edited the manuscript. E.W. and J.K. provided resources.

## Competing interests

The authors declare no competing interests.
