## [Peer Review File · Communications Biology]

Reviewers' comments:

Reviewer #1 (Remarks to the Author):

Lardon et al. described the GWAS analysis on shoot regeneration traits using 190 natural accessions of *Arabidopsis thaliana*. They found important intersections to influence shoot regeneration traits; the detailed analysis on intersections showed clearly that the critical factors affecting shoot regeneration for all culture conditions/accessions are not so many, ~10 genes would be considered as such "universal master regulators of shoot regeneration", and the majority of QTGs here would be "condition-specific regulars of shoot regeneration". In addition, the authors revealed the benefit SNPs in specific genes for shoot regeneration. One of important genes for shoot regeneration with benefit SNPs is WUS, and the expression level of WUS is critical for shoot regeneration traits, as shown previously. The T-DNA insertion analysis indicated that some of T-DNA insertion mutants for newly identified genes from GWAS analysis showed the increased or decreased shoot regeneration efficiency. Together, the data here provided important insights into the complex genetic frame of shoot regeneration.

My major concern on this manuscript is "novel and strong points over previous GWAS works, such as Motte et al. 2014, are unclear". WUS is a well known critical factor for shoot regeneration (but you used 2 figures and 1 table to indicate the importance of WUS), and the half of accessions you used seemed to fail to regenerate shoots, i.e. not greatly contributed to QTL/QTG results, I guess. The T-DNA insertion lines showed the different response of mutants to different culture condition, but you did not show the control experiment (I mean, to check shoot regeneration of randomly selected T-DNA mutants). Thus I am not sure that this results are significantly positive or not as methodology, because the results are not drastic actually.

Other points;

1. Discussion; I felt the current discussion is too long and like a kind of rough reviews on the related genes. Of course the basic knowledge on such genes should be introduced, but the authors should discuss the points that you could improve our understanding of shoot regeneration based on your data; how communicate such genes for shoot regeneration in response to different culture conditions? Which aspects are first findings to compare with other GWAS-shoot regeneration works (including other plant species than *Arabidopsis*)? Such discussion should be more important than to mention the genes you found one by one. (The authors showed Figure 6, but not so much discussed within this interesting figure.)

2. Figure 2 is important data for your manuscript, but it is tough to understand this figure. Which intersections contain what kinds of genes in Fig 2D and 2E? For Fig 2B, the SNPs of interest with significant p values can be marked for easy understanding. The data shown in Fig 2F is not well explained/used in the main text.

3. I could not find Dataset 5 and 6, as well as SI Materials & Methods, in reviewing system. Anyway for understanding your data, the shoot regeneration traits used for GWAS analysis and the criteria of selected genes for detailed analysis are very important. These data would be main data, not supplementary.

Reviewer #2 (Remarks to the Author):

While recent studies have begun to elucidate underlying cellular and molecular mechanisms of de novo shoot regeneration, the genetic basis of variations in regeneration efficiency across the plant species remain poorly understood. Manuscript by Lardon et al., describe in depth genetic analysis of natural variation in regeneration efficiency across various *Arabidopsis* accessions. The authors show circumstantial correlation between WUSCHEL expression and variations in regeneration

efficiency of wildtype accessions using molecular-genetic approaches and bioinformatics. The authors link allelic variation in cis regulatory sequences of WUS to variations in shoot regeneration. They have performed a comprehensive and meticulous analysis of the tested parameters under a variety of in vitro conditions. Though the study provides comprehensive and carefully done analysis, it requires substantial revision.

A burst of recent studies have begun to unravel the mechanism underlying acquisition of pluripotent state in regenerative mass called callus and subsequent de novo shoot regeneration. Authors should provide more comprehensive introduction to make readers familiar with the work done in this area.

68-69: Transdifferentiation refers to the direct conversion of one fate to another. Though callus on CIM follow the root pathway and thus can be considered as a transdifferentiating mass, literature from last five years shows that root morphology is lost upon incubation on shoot induction medium (SIM). Shoot is formed only during shoot induction phase on SIM. Therefore statements such as "root like protuberances to shoot primordia" is misleading. Reference (Rosspopoff et al) cited is not appropriate for indirect shoot regeneration as it discusses only direct shoot organogenesis which does not involve any callus formation.

Line 103-105: Inferring that failure to regenerate is because of 'a handful of factors' in the beginning of the results section is not appropriate. The authors have appropriately concluded this result towards the end of the manuscript.

Line 115: Somaclonal variation is unlikely to be accounted for variation in regeneration efficiency. Figure1: It would be useful to give distinct colours to accessions having different regeneration capacity as the brown and green on the extreme ends of the key are not clearly distinguishable on the map.

Line 211: What was the criterion for performing qRT PCR for measuring WUS transcript levels upon 3day incubation on SIM. Often confinement of WUS in shoot foci and further development occurs much later during SIM incubation.

Figure4: WUS is expressed in the promeristem much prior to the formation of leaf primordia/shoot. WUS regulated stem cell activity contribute to leaf primordia formation. It is not understood why WUS (located on chromosome2) is not required for primordia formation (linked to chromosome1).

Figure4: How the shoot primordia are quantified is unclear from the methods section. The large error bar may be due to variation in the number of primordia due to progressive shoot development stages or because of incorrect criteria of scoring (green foci misinterpreted as primordia). Green foci are only sites of chlorophyll maturation and they do not necessarily harbour any sign of productive shoot formation, not even any molecular marker related to shoot regeneration. Therefore using green foci as a criteria will be misleading. Not surprisingly the authors detect huge variations while using such criteria. The authors can restrict the quantification to number of shoot or preferably choose an unambiguous molecular criterion. But at present the authors may not need to provide any molecular criteria for this work.

Figure4: what does the green and cream colour highlights in chromosome pattern depict? Add this information to figure legend.

line 223-227: When the expression of WUS has been interpreted to be increasing does the authors mean that the number of WUS expressing shoot foci has increased and therefore they observe increased shoot formation or whether the expression of WUS has increased within individual foci? Further studies would be required to understand the pattern of WUS expression, however the authors could evoke these possibilities in the discussion section.

The authors have demonstrated that there is a striking correlation between allelic variation in WUS and the variation in shoot regeneration potential. Variation in regeneration is an outcome of various factors. The authors need be open to the possibility that the enhancement in shoot regeneration could be a result of multiple regulatory inputs including those that originated during acquisition of callus pluripotency.

Following CSL the improvement in regeneration need not necessarily be only because of 3 fold increase in WUS transcript levels. From previous literature it is known that substantial level of WUS is required to increase the regeneration. Having said that, 3 fold increase in WUS transcript may not be the only factor. The authors need to take this into account.

248-250: Often mutation in only one gene may not show defect in the biological process under investigation due to extreme redundancy. Also T-DNA insertions may not generate completely null mutants. Please consider these facts while revising the results.

Figure5: How is the regenerated shoot 'area' measured? Does it include greening of the callus or only shoot and 'primordia' formation. As mentioned earlier, greening should not be considered as an indication of shoot regeneration as it only depicts chlorophyll maturation. It would be useful for the readers if stereomicroscope micrographs (representing shoot primordia/ root like structures/ undefined structures) are provided in the manuscript. The bar representing "protocol a" for at1g20380 is thicker than the rest, please use uniform thickness for the bars.

325: Instead of using the term totipotent, restrict to the term pluripotency as a complete bipolar plant with shoot and root poles are not produced during de novo shoot organogenesis. Only on subsequent exposure to root induction medium, these shoots generate root.

I was wondering why epigenetic regulators and cell cycle regulators/inhibitors were not detected in these analysis of various accessions. Atleast the authors should discuss this in the discussion.

While recent studies have begun to elucidate underlying cellular and molecular mechanisms of *de novo* shoot regeneration, the genetic basis of variations in regeneration efficiency across the plant species remain poorly understood. Manuscript by Lardon *et al.*, describe in depth genetic analysis of natural variation in regeneration efficiency across various Arabidopsis accessions. The authors show circumstantial correlation between WUSCHEL expression and variations in regeneration efficiency of wildtype accessions using molecular-genetic approaches and bioinformatics. The authors link allelic variation in *cis* regulatory sequences of WUS to variations in shoot regeneration. They have performed a comprehensive and meticulous analysis of the tested parameters under a variety of *in vitro* conditions. Though the study provides comprehensive and carefully done analysis, it requires substantial revision.

1. A burst of recent studies have begun to unravel the mechanism underlying acquisition of pluripotent state in regenerative mass called callus and subsequent *de novo* shoot regeneration. Authors should provide more comprehensive introduction to make readers familiar with the work done in this area.
2. 68-69: Transdifferentiation refers to the direct conversion of one fate to another. Though callus on CIM follow the root pathway and thus can be considered as a transdifferentiating mass, literature from last five years shows that root morphology is lost upon incubation on shoot induction medium (SIM). Shoot is formed only during shoot induction phase on SIM. Therefore statements such as “root like protuberances to shoot primordia” is misleading. Reference (Rosspopoff et al) cited is not appropriate for indirect shoot regeneration as it discusses only direct shoot organogenesis which does not involve any callus formation.
3. Line 103-105: Inferring that failure to regenerate is because of ‘a handful of factors’ in the beginning of the results section is not appropriate. The authors have appropriately concluded this result towards the end of the manuscript.
4. Line 115: Somaclonal variation is unlikely to be accounted for variation in regeneration efficiency.
5. Figure1: It would be useful to give distinct colours to accessions having different regeneration capacity as the brown and green on the extreme ends of the key are not clearly distinguishable on the map.
6. Line 211: What was the criterion for performing qRT PCR for measuring WUS transcript levels upon 3day incubation on SIM. Often confinement of WUS in shoot foci and further development occurs much later during SIM incubation.
7. Figure4: WUS is expressed in the promeristem much prior to the formation of leaf primordia/shoot. WUS regulated stem cell activity contribute to leaf primordia formation. It is not understood why WUS (located on chromosome2) is not required for primordia formation (linked to chromosome1).
8. Figure4: How the shoot primordia are quantified is unclear from the methods section. The large error bar may be due to variation in the number of primordia due to progressive shoot development stages or because of incorrect criteria of scoring (green foci misinterpreted as primordia). Green foci are only sites of chlorophyll maturation and they do not necessarily harbour any sign of productive shoot formation, not even any molecular marker related to shoot regeneration. Therefore using green foci as a criteria will be misleading. Not surprisingly the authors detect huge variations while using such criteria. The authors can restrict the quantification to number of shoot or preferably choose an unambiguous molecular criterion. But at present the authors may not need to provide any molecular criteria for this work.
9. Figure4: what does the green and cream colour highlights in chromosome pattern depict? Add this information to figure legend.

10. line 223-227: When the expression of WUS has been interpreted to be increasing does the authors mean that the number of WUS expressing shoot foci has increased and therefore they observe increased shoot formation or whether the expression of WUS has increased within individual foci? Further studies would be required to understand the pattern of WUS expression, however the authors could evoke these possibilities in the discussion section.
11. The authors have demonstrated that there is a striking correlation between allelic variation in WUS and the variation in shoot regeneration potential. Variation in regeneration is an outcome of various factors. The authors need be open to the possibility that the enhancement in shoot regeneration could be a result of multiple regulatory inputs including those that originated during acquisition of callus pluripotency.
12. Following CSL the improvement in regeneration need not necessarily be only because of 3 fold increase in WUS transcript levels. From previous literature it is known that substantial level of WUS is required to increase the regeneration. Having said that, 3 fold increase in WUS transcript may not be the only factor. The authors need to take this into account.
13. 248-250: Often mutation in only one gene may not show defect in the biological process under investigation due to extreme redundancy. Also T-DNA insertions may not generate completely null mutants. Please consider these facts while revising the results.
14. Figure5: How is the regenerated shoot 'area' measured? Does it include greening of the callus or only shoot and 'primordia' formation. As mentioned earlier, greening should not be considered as an indication of shoot regeneration as it only depicts chlorophyll maturation. It would be useful for the readers if stereomicroscope micrographs (representing shoot primordia/ root like structures/ undefined structures) are provided in the manuscript. The bar representing "protocol a" for at1g20380 is thicker than the rest, please use uniform thickness for the bars.
15. 325: Instead of using the term totipotent, restrict to the term pluripotency as a complete bipolar plant with shoot and root poles are not produced during *de novo* shoot organogenesis. Only on subsequent exposure to root induction medium, these shoots generate root.
16. I was wondering why epigenetic regulators and cell cycle regulators/inhibitors were not detected in these analysis of various accessions. Atleast the authors should discuss this in the discussion.

Reviewer #1 (Remarks to the Author):

Lardon et al. described the GWAS analysis on shoot regeneration traits using 190 natural accessions of *Arabidopsis thaliana*. They found important intersections to influence shoot regeneration traits; the detailed analysis on intersections showed clearly that the critical factors affecting shoot regeneration for all culture conditions/accessions are not so many, ~10 genes would be considered as such “universal master regulators of shoot regeneration”, and the majority of QTGs here would be “condition-specific regulars of shoot regeneration”. In addition, the authors revealed the benefit SNPs in specific genes for shoot regeneration. One of important genes for shoot regeneration with benefit SNPs is WUS, and the expression level of WUS is critical for shoot regeneration traits, as shown previously. The T-DNA insertion analysis indicated that some of T-DNA insertion mutants for newly identified genes from GWAS analysis showed the increased or decreased shoot regeneration efficiency. Together, the data here provided important insights into the complex genetic frame of shoot regeneration.

1. My major concern on this manuscript is “novel and strong points over previous GWAS works, such as Motte et al. 2014, are unclear”.
 - ⇒ The present study differs from the work of Motte et al. 2014 in 3 major ways. First, the latter is not a genome-wide association study per definition, as it merely exploited local association analyses to refine 1 of 5 QTLs obtained by linkage mapping using RILs between accessions Ga-0 and Nok-3. In other words, natural variation in regeneration was only correlated to SNPs (from a 250k SNP array) between Ga-0 and Nok-3 in the 1Mb region flanking the f5a1859436 marker on chromosome 1. On top of that, only 88 accessions were phenotyped (and sequencing data was only available for 62 of those), whereas we investigated 190 accessions and retained 149 for association mapping based on whole genome sequences. Note that this is also substantially more than the 48 accessions tested for linkage disequilibrium by Zhang et al. 2018. Lastly, we have investigated more regeneration traits under different conditions (protocol a and b), making the current study more robust than others, which is confirmed by the association near WUSCHEL that has not previously been detected.
2. WUS is a well known critical factor for shoot regeneration (but you used 2 figures and 1 table to indicate the importance of WUS), and the half of accessions you used seemed to fail to regenerate shoots, i.e. not greatly contributed to QTL/QTG results, I guess.
 - ⇒ The role of WUS in regeneration is indeed well-established, but it has not been linked to natural variation before. We wanted to show accurate data for this observation and illustrate how WUS underlies regeneration in multiple conditions, distinguishing it from other typical regeneration genes such as STM. Besides, only one figure is specifically dedicated to WUS, the other also contain information about other QTGs. Regarding poorly regenerating accessions, we believe that these do contribute to proper identification of QTGs, as they allow to filter out those alleles that are shared with strong regenerators and do not contribute to regeneration.
3. The T-DNA insertion lines showed the different response of mutants to different culture condition, but you did not show the control experiment (I mean, to check shoot regeneration of randomly selected T-DNA mutants). Thus I am not sure that this results are significantly positive or not as methodology, because the results are not drastic actually.
 - ⇒ As a negative control, we have tested single mutants for 9 genes that are not linked to the GWAS: *AT1G48820* (a terpenoid cyclase), *AT3G13990* (a dentin sialophosphoprotein), *FH1* (formin homology 1), *THE1* (theseus 1; a receptor kinase with a role in cell elongation), *BIN2* (brassinosteroid-insensitive 2), *COP1* (constitutive photomorphogenic 1), *PIF1* (phytochrome interacting factor 1), *PHYA* (phytochrome A) and *PP2A* (protein phosphatase 2A). The graph below shows that few defects were recorded in these lines and they only persisted across

protocols in *cop1-5* and *pp2aB'--β+--*. Because some of the above genes are involved in light signalling and phosphorylation and these processes are important for regeneration (Nameth et al. 2013, Pulianmackal et al. 2014), we also tested corresponding multiple gene knockouts as a positive control: *phot1-5 phot2* (phototropin 1 and 2), *phyA phyB cry1 cry2* (phytochrome A and B; cryptochrome 1 and 2), *phyA phyB*, *phyA cry1 cry2* and *spa1-7 spa2-1 spa3-1* (suppressor of PHYA 1, 2 and 3). This revealed that while single mutations in critical pathways generally yield little regeneration defects, disruption of multiple homologs often impairs *de novo* organogenesis. Mutants for GWAS candidates showed intermediate phenotypes compared to the negative and positive controls we provide here, but since they all contained single insertions and many of these genes act redundantly (see discussion), we conclude that the observed effects were biologically relevant. Finally, we note that some lines in Fig. 5 (e.g. *at4g08630*, *egret* and *ubc28*) did not show significant differences to the wild type, confirming that the results are unlikely to be biased by off-target effects or general weakening of the mutants. The figure below has been added to the supplementary information (Fig. S5) and a brief conclusion of this experiment was introduced in the manuscript (line 254-257): “For comparison, we tested 9 unrelated T-DNA mutants and 5 multiple gene knockouts linked to light signalling and phosphorylation, revealing that while the insertions in Fig. 5 are not as detrimental as higher order mutations, they yield more severe defects than random single gene disruptions (Fig. S5).”

Regeneration in random T-DNA insertion lines and mutants related to light signalling and phosphorylation as negative and positive controls for the data presented in Fig. 5.

Other points;

- Discussion; I felt the current discussion is too long and like a kind of rough reviews on the related genes. Of course the basic knowledge on such genes should be introduced, but the authors should discuss the points that you could improve our understanding of shoot regeneration based on your data; how communicate such genes for shoot regeneration in response to different culture conditions? Which aspects are first findings to compare with other GWAS-shoot regeneration works (including other plant species than Arabidopsis)? Such discussion should be more important than to mention the genes you found one by one. (The authors showed Figure 6, but not so much discussed within this interesting figure.)
⇒ The primary goal of this study was to identify (novel) regeneration determinants and because association mapping provides few mechanistic insights into gene function and interactions,

we reviewed literature on the most important candidates to assess whether and how they might play a role. We drew as much connections between the genes as our results and the state of the art allowed and made a functional classification into prior candidates involved in shoot organogenesis and hormone responses, homologues of prior candidates and lesser known candidates. This is visualized in Fig. 6, serving as a guideline for the discussion. To address the other remarks, we shortened the discussion of individual candidates (line 321-414) and compared our findings to similar GWA studies in other species (line 297-307): “a GWAS on embryonic callus regeneration in maize showed that only 15 out of 63 QTNs were retained in multiple environments and highlighted *WUSCHEL-RELATED HOMEODOMAIN 2* (*WOX2*), although other candidates are distinct from ours². Most QTGs we identified are also new compared to association studies on adventitious shoot regeneration in roses³, callus formation in poplar⁴ and rice⁵ and *in vitro* regeneration of cucumber⁶ and tomato⁷. However, these studies do report similar functional classes of candidates (e.g. embryogenesis and meristem genes, reprogramming factors, hormone-related proteins, receptor-like kinases and TFs from the LBD, ERF, MYB and WOX families)²⁻⁵ and in cases where multiple traits, protocols or techniques are evaluated, overlap between them is limited^{2,3}, suggesting that the difference in experimental systems could be part of the cause”.

5. A) Figure 2 is important data for your manuscript, but it is tough to understand this figure. Which intersections contain what kinds of genes in Fig 2D and 2E?
 - ⇒ Dataset 5 contains an excel sheet showing which genes are found in which intersections and the manuscript mentions details of genes in the highest order intersections. For lower order intersections containing many genes, GO enrichment showed no significant or meaningful overrepresentations and we considered it too descriptive to discuss all functional categories in a set (especially as these are just candidate QTGs and the results could be biased by false positives).
 - B) For Fig 2B, the SNPs of interest with significant p values can be marked for easy understanding.
 - ⇒ Unfortunately, the easyGWAS software does not allow to colour SNPs or shift the position of the green bar reflecting the significance threshold. However, the vertical axis in Fig 2B-C is the negative logarithm of the p-value and since the figure legend clearly states that SNPs with $p < 1e-5$ were considered significant in the other panels, colouring the SNPs would provide no additional information (all SNPs in panel B and C above a horizontal line crossing the y axis at a value of 5 would be coloured).
 - C) The data shown in Fig 2F is not well explained/used in the main text.
 - ⇒ A more elaborate interpretation of Fig. 2E-F has been added to the manuscript (line 158-161): “Intriguingly, around 10 factors are linked to many phenotypes across protocols and genes in highlighted intersections are supported by larger SNP clusters with low p-values (Fig. 2E). Moreover, their positive alleles are rare (low MAF) and often correspond to beta values at the edge of the distribution, meaning they contribute substantially to regenerative variation (Fig. 2F)”.
6. I could not find Dataset 5 and 6, as well as SI Materials & Methods, in reviewing system. Anyway for understanding your data, the shoot regeneration traits used for GWAS analysis and the criteria of selected genes for detailed analysis are very important. These data would be main data, not supplementary.
 - ⇒ These are indeed valuable data, but due to size restrictions it is not possible to present them in the manuscript in the form of a table. Therefore, we decided to add them as supplementary datasets, ensuring access for everyone (the file was submitted with the first version of the article). Furthermore, phenotypic data and association results will respectively be submitted to the AraPheno database and the AraGWAS catalogue (accession numbers will be provided at the time of publication).

Reviewer #2 (Remarks to the Author):

While recent studies have begun to elucidate underlying cellular and molecular mechanisms of de novo shoot regeneration, the genetic basis of variations in regeneration efficiency across the plant species remain poorly understood. Manuscript by Lardon et al., describe in depth genetic analysis of natural variation in regeneration efficiency across various Arabidopsis accessions. The authors show circumstantial correlation between WUSCHEL expression and variations in regeneration efficiency of wildtype accessions using molecular-genetic approaches and bioinformatics. The authors link allelic variation in cis regulatory sequences of WUS to variations in shoot regeneration. They have performed a comprehensive and meticulous analysis of the tested parameters under a variety of in vitro conditions. Though the study provides comprehensive and carefully done analysis, it requires substantial revision.

1. A burst of recent studies have begun to unravel the mechanism underlying acquisition of pluripotent state in regenerative mass called callus and subsequent de novo shoot regeneration. Authors should provide more comprehensive introduction to make readers familiar with the work done in this area.
 - ⇒ We tried to keep the introduction brief and instead elaborate on literature specific to our candidates in the discussion, but several key molecular modules behind callus formation, pluripotency acquisition and shoot formation have been added to the introduction (line 61-67 and 82-83): “The convergence of hormone signals (e.g. auxin-induced PLT3, 5 & 7/CUP-SHAPED COTYLEDON (CUC) 1 & 2 and WOX11/LATERAL ORGAN BOUNDARIES DOMAIN (LBD) 16 modules^{8,9}) with stress and wounding responses (e.g. mediated by WOUND-INDUCED DEDIFFERENTIATION (WIND) 1¹⁰) on CIM also underlies the acquisition of competence to regenerate shoots later on¹¹, by reactivating the cell cycle and installing progressive epigenetic changes such as DNA demethylation and histone modifications (e.g. H3K4me2 and H3K27me3)¹²⁻¹⁵” and “PHB, PHV and REV also promote expression of the shoot determinants STM and RAP2.6L and WIND1 contributes to the events on SIM by directly activating ESR1^{11,16}”.
2. 68-69: Transdifferentiation refers to the direct conversion of one fate to another. Though callus on CIM follow the root pathway and thus can be considered as a transdifferentiating mass, literature from last five years shows that root morphology is lost upon incubation on shoot induction medium (SIM). Shoot is formed only during shoot induction phase on SIM. Therefore statements such as “root like protuberances to shoot primordia” is misleading. Reference (Rosspopoff et al) cited is not appropriate for indirect shoot regeneration as it discusses only direct shoot organogenesis which does not involve any callus formation.
 - ⇒ Many studies on indirect shoot regeneration have also described the process as a transdifferentiation from lateral root-like promeristems to shoot apical meristems rather than a process of true cell de-differentiation (Atta et al 2008, Sugimoto et al. 2010 & 2011). Of course, this implies that root fate is lost on SIM, but I found no recent literature stating that this compromises the idea of direct conversion. Rosspopoff et al. showed that root primordia can directly be transformed into shoots and provided details of the developmental window in which this occurs. Although it is indeed a study of direct organogenesis, similar time restrictions have been reported for callus formation and competence acquisition (Cary et al. 2002, Gordon et al. 2007, Pulianmackal et al. 2014). References for this part have been updated in the manuscript (line 73).
3. Line 103-105: Inferring that failure to regenerate is because of ‘a handful of factors’ in the beginning of the results section is not appropriate. The authors have appropriately concluded this result towards the end of the manuscript.

- ⇒ The manuscript has been updated accordingly (line 108-110): “This multitude of levels in the phenotype can only be explained by numerous small allelic contributions, which suggests that *de novo* shoot organogenesis is a multigenic trait, a notion that agrees with the state of the art¹¹”.
4. Line 115: Somaclonal variation is unlikely to be accounted for variation in regeneration efficiency.
 ⇒ This referred to epigenetic differences that can also be at the base of somaclonal variation, but the formulation was indeed wrong and has been revised (line 118): “variability within accessions is likely due to environmental fluctuations and epigenetic effects”.
5. Figure1: It would be useful to give distinct colours to accessions having different regeneration capacity as the brown and green on the extreme ends of the key are not clearly distinguishable on the map.
 ⇒ The colours in figure 1 have been adjusted (see below).

Figure 1 with updated colours on the geographic distribution (upper panel).

6. Line 211: What was the criterion for performing qRT PCR for measuring WUS transcript levels upon 3day incubation on SIM. Often confinement of WUS in shoot focii and further development occurs much later during SIM incubation.
 ⇒ This timing was chosen as the earliest onset of WUS expression, prior to morphological changes, based on literature (e.g. Atta et al. 2009, Zhang et al. 2017) and because strong differences were recorded, no further time points were investigated.
7. Figure4: WUS is expressed in the promeristem much prior to the formation of leaf primordia/shoot.WUS regulated stem cell activity contribute to leaf primordia formation.It is not understood why WUS (located on chromosome2) is not required for primordia formation (linked to chromosome1).

⇒ WUS might not underlie the difference in regenerated shoot primordia between Col-0 and Ler here (potentially because their low regeneration rate hampers accurate comparison), but significant associations do show up in Manhattan plots for shoot primordia after 15 days under protocol a and b (look at the peaks around ~8 Mbp in the figures below). Therefore, according to GWAS, WUS is linked to the formation of shoot primordia as well.

Manhattan plots for regenerated primordium numbers after 15 days under protocol a and b (upper and lower panel respectively). Associations near WUS are highlighted with a blue circle.

8. Figure4: How the shoot primordia are quantified is unclear from the methods section. The large error bar may be due to variation in the number of primordia due to progressive shoot development stages or because of incorrect criteria of scoring (green foci misinterpreted as primordia). Green foci are only sites of chlorophyll maturation and they do not necessarily harbour any sign of productive shoot formation, not even any molecular marker related to shoot regeneration. Therefore using green foci as a criteria will be misleading. Not surprisingly the authors detect huge variations while using such criteria. The authors can restrict the quantification to number of shoot or preferably choose an unambiguous molecular criterion. But at present the authors may not need to provide any molecular criteria for this work.

⇒ As stated in the methods section, different structures were scored by counting (which was done meticulously and consistently). Shoot primordia were recognized as dome-shaped outgrowths with purple or green colour and a clearly organized cellular patterning, indicated by a smooth surface (cfr. Motte et al. 2014). To clarify how the features were distinguished, representative images (see figure below) have been added to the supplementary information as Fig. S1.

Figure S1: Representative images of the different structures that were scored in 3 variable accessions. Green, blue, red and grey arrows respectively indicate shoots, shoot primordia, root-like structures and undefined structures.

9. Figure4: what does the green and cream colour highlights in chromosome pattern depict? Add this information to figure legend.
- ⇒ These colours show the difference between Col-0 and Ler alleles, in concordance with the legend of the left panel. The caption has been updated (line 243): “The right panel shows the chromosome patterns of the CSLs (with Col-0 alleles in khaki and Ler variants in green)”.
10. line 223-227: When the expression of WUS has been interpreted to be increasing does the authors mean that the number of WUS expressing shoot foci has increased and therefore they observe increased shoot formation or whether the expression of WUS has increased within individual foci? Further studies would be required to understand the pattern of WUS expression, however the authors could evoke these possibilities in the discussion section.
- ⇒ This is a very relevant question, but at this point we do not have data supporting further conclusions on the matter.
11. The authors have demonstrated that there is a striking correlation between allelic variation in WUS and the variation in shoot regeneration potential. Variation in regeneration is an outcome of various factors. The authors need be open to the possibility that the enhancement in shoot regeneration could be a result of multiple regulatory inputs including those that originated during acquisition of callus pluripotency.
- ⇒ The notion that WUS is not the only determinant of variation in regeneration and that regulation may differ at various stages has been stressed more clearly in the manuscript (line 205-207): “Nonetheless, our GWAS shows that various other factors contribute to the observed variability, which is likely a result of differential regulation at various stages of *de novo* shoot organogenesis, including founder cell specification, pluripotency acquisition and SAM patterning”.
12. Following CSL the improvement in regeneration need not necessarily be only because of 3 fold increase in WUS transcript levels. From previous literature it is known that substantial level of WUS is required to increase the regeneration. Having said that, 3 fold increase in WUS transcript may not be the only factor. The authors need to take this into account.
- ⇒ The manuscript has been updated accordingly (line 230-234): “Chromosome 1 appears to be important as well, because lines with a Col-0 variant form more shoot primordia than those with the Ler version (Fig. 4; left panel) and significant interactions were found between chr3:chr5, chr1:chr4 and chr2:chr4. Together with the small regenerative difference between Col-0 and Ler, this suggests that WUS is not the only factor at play and variation between these accessions is orchestrated by a combination of positive and negative inputs”.
13. 248-250: Often mutation in only one gene may not show defect in the biological process under investigation due to extreme redundancy. Also T-DNA insertions may not generate completely null mutants. Please consider these facts while revising the results.
- ⇒ The manuscript has been updated accordingly (line 254 and 262-264): “none of the lines completely lost their regenerative capacity, potentially because of gene redundancy or incomplete loss-of-function” and “effects of individual T-DNA insertions are also small compared to variation between protocols and although this could again be attributed to redundancy or weak null alleles, it suggests that single gene contributions are subordinate to environmental changes”.
14. Figure5: How is the regenerated shoot ‘area’ measured? Does it include greening of the callus or only shoot and ‘primordia’ formation. As mentioned earlier, greening should not be considered as an indication of shoot regeneration as it only depicts chlorophyll maturation. It would be useful for the readers if stereomicroscope micrographs (representing shoot primordia/ root like structures/ undefined structures) are provided in the manuscript. The bar representing “protocol a” for at1g20380 is thicker than the rest, please use uniform thickness for the bars.

⇒ The regenerated area is determined by setting a colour threshold to distinguish explants from the background and subtracting the latter to create a selection, of which the area is calculated. Hence, it reflects a 2D projection of entire explants (rather than the greenness) and the figures below illustrate that this works well as a proxy for regenerated shoot numbers in our system (as shoot numbers are significantly correlated the area). We applied this strategy to quantify regeneration in the T-DNA lines instead of counting, because wild type Col-0 forms very few shoots and this makes it hard to detect reduced regeneration rates in the mutants. The thickness of the bars has been adjusted, to make sure missing values are not interpreted as 0, extra info was added to the caption (line 274-275): “No data was available for *at1g20380* under protocol c and *wavh2* under protocol a”.

Correlation between all phenotypes scored in the GWAS (left) and comparison of regenerated shoot numbers with regenerated green area in 150 accessions (right).

Figure 5 with updated bar thickness.

15. 325: Instead of using the term totipotent, restrict to the term pluripotency as a complete bipolar plant with shoot and root poles are not produced during de novo shoot organogenesis. Only on subsequent exposure to root induction medium, these shoots generate root.

⇒ This is the terminology used by Qiao et al. 2012 to distinguish their C1 and C2 calli and it does not refer to our regeneration assay.

16. I was wondering why epigenetic regulators and cell cycle regulators/inhibitors were not detected in these analysis of various accessions. Atleast the authors should discuss this in the discussion.
- ⇒ We did find a number of epigenetic/cell cycle regulators, such as LDL2, E2FB, AT1G20290 (SWI-SNF-related chromatin-binding protein), VRN2 and HDA10. Possible reasons why other known regeneration genes, including cell cycle genes and epigenetic factors, were not found have been added to the discussion (line 307-314): “Notably, several established SAM genes, epigenetic factors and cell cycle regulators (e.g. STM, CUCs, ESRs, PLTs, WIND1, MET1 and CYCD3¹⁷) were not detected in our assay, which might be due to a lack of functional sequence variation at these loci in the tested population¹⁸. In turn, this could be the result of stringent selection against harmful mutations in genes that are vital to embryonic development, wound repair and rooting. Possibly, epigenetic, transcriptional or post-translational regulation is favoured for key survival genes to allow for better fine-tuning. Investigating the role of these mechanisms in natural regenerative variability by means of eQTL mapping and methylome-wide associations is a promising future prospect¹⁹”.

REVIEWERS' COMMENTS:

Reviewer #1 (Remarks to the Author):

The authors responded to my requests sincerely, and I felt the manuscript has been improved to show new information on shoot regeneration-related genes based on GWAS analysis. Here I would like to ask to revise several points again.

1. For my previous Comment 1 (novel and strong points over previous GWAS works, such as Motte et al. 2014, are unclear);

I understand the answers by authors, and of course I have agreed with the view that the increased samples should expand our understanding of molecular mechanisms for specific traits greatly; but if so, could you please briefly mention such stronger (or different) points to compare with the previous ones, in the beginning of Discussion? You started Discussion within "As previously reported," in the current manuscript. This made readers confused to understand how much the findings here is new.

2. For my previous Comment 5B (For Fig 2B, the SNPs of interest with significant p values can be marked for easy understanding);

Your answer was "Unfortunately, the easyGWAS software does not allow to colour SNPs or shift the position of the green bar reflecting the significance threshold." But I think anyway you can add red circles or something by yourself, to indicate which dots are mentioned to be significant in main text. Honestly, Figure 2 is so crowded with information and each panel is small, thus it is hard to find out the points you mentioned.

3. For supplemental datasets; this time I can get the files successfully. However, it was tough to recognize which tables contain what kinds of data, since you didn't put any Table titles nor labels. Could you please put the title for each sheet?

As well, I could not find the file to describe the full name of 190 accessions and their location exactly. Panels of Figure 1 are too small and almost impossible to recognize what you used correctly (because you used only abbreviations in the main manuscript). Please add such crucial information on your work.

4. For Ler; "er" should be italic.

Reviewer #2 (Remarks to the Author):

The authors have addressed my comments adequately. The revised manuscript reads well.

I only have a minor point. I suggest authors to add the following in the discussion:

At present we can not distinguish between the two possibilities whether number of WUS expressing shoot foci has increased and therefore we observe increased shoot formation or whether the expression of WUS has increased within individual foci.

First round of revision

Reviewer #1 (Remarks to the Author):

Lardon et al. described the GWAS analysis on shoot regeneration traits using 190 natural accessions of *Arabidopsis thaliana*. They found important intersections to influence shoot regeneration traits; the detailed analysis on intersections showed clearly that the critical factors affecting shoot regeneration for all culture conditions/accessions are not so many, ~10 genes would be considered as such “universal master regulators of shoot regeneration”, and the majority of QTGs here would be “condition-specific regulars of shoot regeneration”. In addition, the authors revealed the benefit SNPs in specific genes for shoot regeneration. One of important genes for shoot regeneration with benefit SNPs is *WUS*, and the expression level of *WUS* is critical for shoot regeneration traits, as shown previously. The T-DNA insertion analysis indicated that some of T-DNA insertion mutants for newly identified genes from GWAS analysis showed the increased or decreased shoot regeneration efficiency. Together, the data here provided important insights into the complex genetic frame of shoot regeneration.

1. My major concern on this manuscript is “novel and strong points over previous GWAS works, such as Motte et al. 2014, are unclear”.
 - ⇒ The present study differs from the work of Motte et al. 2014 in 3 major ways. First, the latter is not a genome-wide association study per definition, as it merely exploited local association analyses to refine 1 of 5 QTLs obtained by linkage mapping using RILs between accessions Ga-0 and Nok-3. In other words, natural variation in regeneration was only correlated to SNPs (from a 250k SNP array) between Ga-0 and Nok-3 in the 1Mb region flanking the f5a1859436 marker on chromosome 1. On top of that, only 88 accessions were phenotyped (and sequencing data was only available for 62 of those), whereas we investigated 190 accessions and retained 149 for association mapping based on whole genome sequences. Note that this is also substantially more than the 48 accessions tested for linkage disequilibrium by Zhang et al. 2018. Lastly, we have investigated more regeneration traits under different conditions (protocol a and b), making the current study more robust than others, which is confirmed by the association near *WUSCHEL* that has not previously been detected.
2. *WUS* is a well known critical factor for shoot regeneration (but you used 2 figures and 1 table to indicate the importance of *WUS*), and the half of accessions you used seemed to fail to regenerate shoots, i.e. not greatly contributed to QTL/QTG results, I guess.
 - ⇒ The role of *WUS* in regeneration is indeed well-established, but it has not been linked to natural variation before. We wanted to show accurate data for this observation and illustrate how *WUS* underlies regeneration in multiple conditions, distinguishing it from other typical regeneration genes such as *STM*. Besides, only one figure is specifically dedicated to *WUS*, the other also contain information about other QTGs. Regarding poorly regenerating accessions, we believe that these do contribute to proper identification of QTGs, as they allow to filter out those alleles that are shared with strong regenerators and do not contribute to regeneration.
3. The T-DNA insertion lines showed the different response of mutants to different culture condition, but you did not show the control experiment (I mean, to check shoot regeneration of randomly selected T-DNA mutants). Thus I am not sure that this results are significantly positive or not as methodology, because the results are not drastic actually.
 - ⇒ As a negative control, we have tested single mutants for 9 genes that are not linked to the GWAS: *AT1G48820* (a terpenoid cyclase), *AT3G13990* (a dentin sialophosphoprotein), *FH1* (formin homology 1), *THE1* (theseus 1; a receptor kinase with a role in cell elongation), *BIN2* (brassinosteroid-insensitive 2), *COP1* (constitutive photomorphogenic 1), *PIF1*

(phytochrome interacting factor 1), *PHYA* (phytochrome A) and *PP2A* (protein phosphatase 2A). The graph below shows that few defects were recorded in these lines and they only persisted across protocols in *cop1-5* and *pp2aB'--β+*. Because some of the above genes are involved in light signalling and phosphorylation and these processes are important for regeneration (Nameth et al. 2013, Pulianmackal et al. 2014), we also tested corresponding multiple gene knockouts as a positive control: *phot1-5 phot2* (phototropin 1 and 2), *phyA phyB cry1 cry2* (phytochrome A and B; cryptochrome 1 and 2), *phyA phyB, phyA cry1 cry2* and *spa1-7 spa2-1 spa3-1* (suppressor of PHYA 1, 2 and 3). This revealed that while single mutations in critical pathways generally yield little regeneration defects, disruption of multiple homologs often impairs *de novo* organogenesis. Mutants for GWAS candidates showed intermediate phenotypes compared to the negative and positive controls we provide here, but since they all contained single insertions and many of these genes act redundantly (see discussion), we conclude that the observed effects were biologically relevant. Finally, we note that some lines in Fig. 7 (e.g. *at4g08630*, *egret* and *ubc28*) did not show significant differences to the wild type, confirming that the results are unlikely to be biased by off-target effects or general weakening of the mutants. The figure below has been added to the Supplementary Information (Supplementary Fig. 5) and a brief conclusion of this experiment was introduced in the manuscript (line 254-257): “For comparison, we tested 9 unrelated T-DNA mutants and 5 multiple gene knockouts linked to light signalling and phosphorylation, revealing that while the insertions in Fig. 7 are not as detrimental as higher order mutations, they yield more severe defects than random single gene disruptions (Supplementary Fig. 5).”

Regeneration in random T-DNA insertion lines and mutants related to light signalling and phosphorylation as negative and positive controls for the data presented in Fig. 7.

Other points;

- Discussion; I felt the current discussion is too long and like a kind of rough reviews on the related genes. Of course the basic knowledge on such genes should be introduced, but the authors should discuss the points that you could improve our understanding of shoot regeneration based on your data; how communicate such genes for shoot regeneration in response to different culture conditions? Which aspects are first findings to compare with other GWAS-shoot regeneration works (including other plant species than Arabidopsis)? Such discussion should be more important than to mention the genes you found one by one. (The authors showed Figure 8, but not so much discussed within this interesting figure.)

⇒ The primary goal of this study was to identify (novel) regeneration determinants and because association mapping provides few mechanistic insights into gene function and interactions, we reviewed literature on the most important candidates to assess whether and how they might play a role. We drew as much connections between the genes as our results and the state of the art allowed and made a functional classification into prior candidates involved in shoot organogenesis and hormone responses, homologues of prior candidates and lesser known candidates. This is visualized in Fig. 8, serving as a guideline for the discussion. To address the other remarks, we shortened the discussion of individual candidates (line 321-414) and compared our findings to similar GWA studies in other species (line 297-307): “a GWAS on embryonic callus regeneration in maize showed that only 15 out of 63 QTNs were retained in multiple environments and highlighted *WUSCHEL-RELATED HOMEODOMAIN 2 (WOX2)*, although other candidates are distinct from ours². Most QTGs we identified are also new compared to association studies on adventitious shoot regeneration in roses³, callus formation in poplar⁴ and rice⁵ and *in vitro* regeneration of cucumber⁶ and tomato⁷. However, these studies do report similar functional classes of candidates (e.g. embryogenesis and meristem genes, reprogramming factors, hormone-related proteins, receptor-like kinases and TFs from the LBD, ERF, MYB and WOX families)²⁻⁵ and in cases where multiple traits, protocols or techniques are evaluated, overlap between them is limited^{2,3}, suggesting that the difference in experimental systems could be part of the cause”.

5. A) Figure 2-3 is important data for your manuscript, but it is tough to understand this figure. Which intersections contain what kinds of genes in Fig 3a and 3b?

⇒ Dataset 5 contains an excel sheet showing which genes are found in which intersections and the manuscript mentions details of genes in the highest order intersections. For lower order intersections containing many genes, GO enrichment showed no significant or meaningful overrepresentations and we considered it too descriptive to discuss all functional categories in a set (especially as these are just candidate QTGs and the results could be biased by false positives).

B) For Fig 2a, the SNPs of interest with significant p values can be marked for easy understanding.

⇒ Unfortunately, the easyGWAS software does not allow to colour SNPs or shift the position of the green bar reflecting the significance threshold. However, the vertical axis in Fig 2a-b is the negative logarithm of the p-value and since the figure legend clearly states that SNPs with $p < 1e-5$ were considered significant in the other panels, colouring the SNPs would provide no additional information (all SNPs in panel B and C above a horizontal line crossing the y axis at a value of 5 would be coloured).

C) The data shown in Fig 3c is not well explained/used in the main text.

⇒ A more elaborate interpretation of Fig. 3b-c has been added to the manuscript (line 158-161): “Intriguingly, around 10 factors are linked to many phenotypes across protocols and genes in highlighted intersections are supported by larger SNP clusters with low p-values (Fig. 3b). Moreover, their positive alleles are rare (low MAF) and often correspond to beta values at the edge of the distribution, meaning they contribute substantially to regenerative variation (Fig. 3c)”.

6. I could not find Dataset 5 and 6, as well as SI Materials & Methods, in reviewing system. Anyway for understanding your data, the shoot regeneration traits used for GWAS analysis and the criteria of selected genes for detailed analysis are very important. These data would be main data, not supplementary.

⇒ These are indeed valuable data, but due to size restrictions it is not possible to present them in the manuscript in the form of a table. Therefore, we decided to add them as supplementary datasets, ensuring access for everyone (the file was submitted with the first

version of the article). Furthermore, phenotypic data and association results will respectively be submitted to the AraPheno database and the AraGWAS catalogue (accession numbers will be provided at the time of publication).

Reviewer #2 (Remarks to the Author):

While recent studies have begun to elucidate underlying cellular and molecular mechanisms of de novo shoot regeneration, the genetic basis of variations in regeneration efficiency across the plant species remain poorly understood. Manuscript by Lardon et al., describe in depth genetic analysis of natural variation in regeneration efficiency across various Arabidopsis accessions. The authors show circumstantial correlation between WUSCHEL expression and variations in regeneration efficiency of wildtype accessions using molecular-genetic approaches and bioinformatics. The authors link allelic variation in cis regulatory sequences of WUS to variations in shoot regeneration. They have performed a comprehensive and meticulous analysis of the tested parameters under a variety of in vitro conditions. Though the study provides comprehensive and carefully done analysis, it requires substantial revision.

1. A burst of recent studies have begun to unravel the mechanism underlying acquisition of pluripotent state in regenerative mass called callus and subsequent de novo shoot regeneration. Authors should provide more comprehensive introduction to make readers familiar with the work done in this area.

⇒ We tried to keep the introduction brief and instead elaborate on literature specific to our candidates in the discussion, but several key molecular modules behind callus formation, pluripotency acquisition and shoot formation have been added to the introduction (line 61-67 and 82-83): “The convergence of hormone signals (e.g. auxin-induced PLT3, 5 & 7/CUP-SHAPED COTYLEDON (CUC) 1 & 2 and WOX11/LATERAL ORGAN BOUNDARIES DOMAIN (LBD) 16 modules^{8,9}) with stress and wounding responses (e.g. mediated by WOUND-INDUCED DEDIFFERENTIATION (WIND) 1¹⁰) on CIM also underlies the acquisition of competence to regenerate shoots later on¹¹, by reactivating the cell cycle and installing progressive epigenetic changes such as DNA demethylation and histone modifications (e.g. H3K4me2 and H3K27me3)¹²⁻¹⁵” and “PHB, PHV and REV also promote expression of the shoot determinants STM and RAP2.6L and WIND1 contributes to the events on SIM by directly activating ESR1^{11,16}”.

2. 68-69: Transdifferentiation refers to the direct conversion of one fate to another. Though callus on CIM follow the root pathway and thus can be considered as a transdifferentiating mass, literature from last five years shows that root morphology is lost upon incubation on shoot induction medium (SIM). Shoot is formed only during shoot induction phase on SIM. Therefore statements such as “root like protuberances to shoot primordia” is misleading. Reference (Rosspopoff et al) cited is not appropriate for indirect shoot regeneration as it discusses only direct shoot organogenesis which does not involve any callus formation.

⇒ Many studies on indirect shoot regeneration have also described the process as a transdifferentiation from lateral root-like promeristems to shoot apical meristems rather than a process of true cell de-differentiation (Atta et al. 2008, Sugimoto et al. 2010 & 2011). Of course, this implies that root fate is lost on SIM, but I found no recent literature stating that this compromises the idea of direct conversion. Rosspopoff et al. showed that root primordia can directly be transformed into shoots and provided details of the developmental window in which this occurs. Although it is indeed a study of direct organogenesis, similar time restrictions have been reported for callus formation and competence acquisition (Cary et al. 2002, Gordon et al. 2007, Pulianmackal et al. 2014). References for this part have been updated in the manuscript (line 73).

3. Line 103-105: Inferring that failure to regenerate is because of ‘a handful of factors’ in the beginning of the results section is not appropriate. The authors have appropriately concluded this result towards the end of the manuscript.
 - ⇒ The manuscript has been updated accordingly (line 108-110): “This multitude of levels in the phenotype can only be explained by numerous small allelic contributions, which suggests that *de novo* shoot organogenesis is a multigenic trait, a notion that agrees with the state of the art¹¹”.
4. Line 115: Somaclonal variation is unlikely to be accounted for variation in regeneration efficiency.
 - ⇒ This referred to epigenetic differences that can also be at the base of somaclonal variation, but the formulation was indeed wrong and has been revised (line 118): “variability within accessions is likely due to environmental fluctuations and epigenetic effects”.
5. Figure1: It would be useful to give distinct colours to accessions having different regeneration capacity as the brown and green on the extreme ends of the key are not clearly distinguishable on the map.
 - ⇒ The colours in figure 1 have been adjusted (see below).

Figure 1 with updated colours on the geographic distribution (upper panel).

6. Line 211: What was the criterion for performing qRT PCR for measuring WUS transcript levels upon 3day incubation on SIM. Often confinement of WUS in shoot foci and further development occurs much later during SIM incubation.
 - ⇒ This timing was chosen as the earliest onset of WUS expression, prior to morphological changes, based on literature (e.g. Atta et al. 2009, Zhang et al. 2017) and because strong differences were recorded, no further time points were investigated.

7. Figure6: WUS is expressed in the promeristem much prior to the formation of leaf primordia/shoot. WUS regulated stem cell activity contribute to leaf primordia formation. It is not understood why WUS (located on chromosome2) is not required for primordia formation (linked to chromosome1).

⇒ WUS might not underlie the difference in regenerated shoot primordia between Col-0 and Ler here (potentially because their low regeneration rate hampers accurate comparison), but significant associations do show up in Manhattan plots for shoot primordia after 15 days under protocol a and b (look at the peaks around ~8 Mbp in the figures below). Therefore, according to GWAS, WUS is linked to the formation of shoot primordia as well.

Manhattan plots for regenerated primordium numbers after 15 days under protocol a and b (upper and lower panel respectively). Associations near WUS are highlighted with a blue circle.

8. Figure6: How the shoot primordia are quantified is unclear from the methods section. The large error bar may be due to variation in the number of primordia due to progressive shoot development stages or because of incorrect criteria of scoring (green foci misinterpreted as primordia). Green foci are only sites of chlorophyll maturation and they do not necessarily harbour any sign of productive shoot formation, not even any molecular marker related to shoot regeneration. Therefore using green foci as a criteria will be misleading. Not surprisingly the authors detect huge variations while using such criteria. The authors can restrict the quantification to number of shoot or preferably choose an unambiguous molecular criterion. But at present the authors may not need to provide any molecular criteria for this work.

⇒ As stated in the methods section, different structures were scored by counting (which was done meticulously and consistently). Shoot primordia were recognized as dome-shaped outgrowths with purple or green colour and clearly organized cellular patterning, indicated by a smooth surface (cfr. Motte et al. 2014). To clarify how the features were distinguished, representative images are provided in Supplementary Fig. 1 (see below).

Supplementary Figure 1: Representative images of the different structures that were scored in 3 variable accessions. Green, blue, red and grey arrows respectively indicate shoots, shoot primordia, root-like structures and undefined structures.

9. Figure6: what does the green and cream colour highlights in chromosome pattern depict? Add this information to figure legend.
 - ⇒ These colours show the difference between Col-0 and Ler alleles, in concordance with the legend of the left panel. The caption has been updated (line 243): “The right panel shows the chromosome patterns of the CSLs (with Col-0 alleles in khaki and Ler variants in green)”.
10. line 223-227: When the expression of WUS has been interpreted to be increasing does the authors mean that the number of WUS expressing shoot foci has increased and therefore they observe increased shoot formation or whether the expression of WUS has increased within individual foci? Further studies would be required to understand the pattern of WUS expression, however the authors could evoke these possibilities in the discussion section.
 - ⇒ This is a very relevant question, but at this point we do not have data supporting further conclusions on the matter.
11. The authors have demonstrated that there is a striking correlation between allelic variation in WUS and the variation in shoot regeneration potential. Variation in regeneration is an outcome of various factors. The authors need be open to the possibility that the enhancement in shoot regeneration could be a result of multiple regulatory inputs including those that originated during acquisition of callus pluripotency.
 - ⇒ The notion that WUS is not the only determinant of variation in regeneration and that regulation may differ at various stages has been stressed more clearly in the manuscript (line 205-207): “Nonetheless, our GWAS shows that various other factors contribute to the observed variability, which is likely a result of differential regulation at various stages of *de novo* shoot organogenesis, including founder cell specification, pluripotency acquisition and SAM patterning”.
12. Following CSL the improvement in regeneration need not necessarily be only because of 3 fold increase in WUS transcript levels. From previous literature it is known that substantial level of WUS is required to increase the regeneration. Having said that, 3 fold increase in WUS transcript may not be the only factor. The authors need to take this into account.
 - ⇒ The manuscript has been updated accordingly (line 230-234): “Chromosome 1 appears to be important as well, because lines with a Col-0 variant form more shoot primordia than those with the Ler version (Fig. 6a) and significant interactions were found between chr3:chr5, chr1:chr4 and chr2:chr4. Together with the small regenerative difference between Col-0 and Ler, this suggests that WUS is not the only factor at play and variation between these accessions is orchestrated by a combination of positive and negative inputs”.
13. 248-250: Often mutation in only one gene may not show defect in the biological process under investigation due to extreme redundancy. Also T-DNA insertions may not generate completely null mutants. Please consider these facts while revising the results.
 - ⇒ The manuscript has been updated accordingly (line 254 and 262-264): “none of the lines completely lost their regenerative capacity, potentially because of gene redundancy or incomplete loss-of-function” and “effects of individual T-DNA insertions are also small compared to variation between protocols and although this could again be attributed to redundancy or weak null alleles, it suggests that single gene contributions are subordinate to environmental changes”.

14. Figure7: How is the regenerated shoot 'area' measured? Does it include greening of the callus or only shoot and 'primordia' formation. As mentioned earlier, greening should not be considered as an indication of shoot regeneration as it only depicts chlorophyll maturation. It would be useful for the readers if stereomicroscope micrographs (representing shoot primordia/ root like structures/ undefined structures) are provided in the manuscript. The bar representing "protocol a" for at1g20380 is thicker than the rest, please use uniform thickness for the bars.

⇒ The regenerated area is determined by setting a colour threshold to distinguish explants from the background and subtracting the latter to create a selection, of which the area is calculated. Hence, it reflects a 2D projection of entire explants (rather than the greenness) and the figures below illustrate that this works well as a proxy for regenerated shoot numbers in our system (as shoot numbers are significantly correlated the area). We applied this strategy to quantify regeneration in the T-DNA lines instead of counting, because wild type Col-0 forms very few shoots and this makes it hard to detect reduced regeneration rates in the mutants. The thickness of the bars has been adjusted, to make sure missing values are not interpreted as 0, extra info was added to the caption (line 274-275): "No data was available for *at1g20380* under protocol c and *wavh2* under protocol a".

Correlation between all phenotypes scored in the GWAS (left) and comparison of regenerated shoot numbers with regenerated green area in 150 accessions (right).

Figure 7 with updated bar thickness.

15. 325: Instead of using the term totipotent, restrict to the term pluripotency as a complete bipolar plant with shoot and root poles are not produced during de novo shoot organogenesis. Only on subsequent exposure to root induction medium, these shoots generate root.
 - ⇒ This is the terminology used by Qiao et al. 2012 to distinguish their C1 and C2 calli and it does not refer to our regeneration assay.
16. I was wondering why epigenetic regulators and cell cycle regulators/inhibitors were not detected in these analysis of various accessions. Atleast the authors should discuss this in the discussion.
 - ⇒ We did find a number of epigenetic/cell cycle regulators, such as LDL2, E2FB, AT1G20290 (SWI-SNF-related chromatin-binding protein), VRN2 and HDA10. Possible reasons why other known regeneration genes, including cell cycle genes and epigenetic factors, were not found have been added to the discussion (line 307-314): “Notably, several established SAM genes, epigenetic factors and cell cycle regulators (e.g. STM, CUCs, ESRs, PLTs, WIND1, MET1 and CYCD3¹⁷) were not detected in our assay, which might be due to a lack of functional sequence variation at these loci in the tested population¹⁸. In turn, this could be the result of stringent selection against harmful mutations in genes that are vital to embryonic development, wound repair and rooting. Possibly, epigenetic, transcriptional or post-translational regulation is favoured for key survival genes to allow for better fine-tuning. Investigating the role of these mechanisms in natural regenerative variability by means of eQTL mapping and methylome-wide associations is a promising future prospect¹⁹”.

Second round of revision

Reviewer #1 (Remarks to the Author):

The authors responded to my requests sincerely, and I felt the manuscript has been improved to show new information on shoot regeneration-related genes based on GWAS analysis.

Here I would like to ask to revise several points again.

1. For my previous Comment 1 (novel and strong points over previous GWAS works, such as Motte et al. 2014, are unclear); I understand the answers by authors, and of course I have agreed with the view that the increased samples should expand our understanding of molecular mechanisms for specific traits greatly; but if so, could you please briefly mention such stronger (or different) points to compare with the previous ones, in the beginning of Discussion? You started Discussion within “As previously reported,” in the current manuscript. This made readers confused to understand how much the findings here is new.
 - ⇒ The discussion has been modified (line 283-285): “In line with previous reports, our extended and genome-wide association analysis found substantial variation in the regenerative potential of 190 *Arabidopsis thaliana* accessions under two different protocols^{28,31}.”
2. For my previous Comment 5B (For Fig 2a-b, the SNPs of interest with significant p values can be marked for easy understanding); Your answer was “Unfortunately, the easyGWAS software does not allow to colour SNPs or shift the position of the green bar reflecting the significance threshold.” But I think anyway you can add red circles or something by yourself, to indicate which dots are mentioned to be significant in main text. Honestly, Figure 2-3 is so crowded with information and each panel is small, thus it is hard to find out the points you mentioned.
 - ⇒ SNPs that were considered significant in the text and subsequent analyses ($p \leq 1e-5$) have been highlighted in red and the figure has been split in two so that panels could be enlarged for better interpretation.

Updated Figure 2 with highlighted SNPs and enlarged panels.

3. For supplemental datasets; this time I can get the files successfully. However, it was tough to recognize which tables contain what kinds of data, since you didn't put any Table titles nor labels. Could you please put the title for each sheet? As well, I could not find the file to describe the full name of 190 accessions and their location exactly. Panels of Figure 1 are too small and almost impossible to recognize what you used correctly (because you used only abbreviations in the main manuscript). Please add such crucial information on your work.
 - ⇒ Titles/descriptions were added to every sheet in the Supplementary Data files, along with references to the corresponding figures. Additional information on the accessions is provided in the source data for Fig. 1b and a new sheet describes the exact locations used in Fig. 1a (note that only 170 accessions were retained in the phenotype plots, because the identity of 20 lines could not be confirmed and only the 150 accessions used for association analyses are shown on the map).
4. For Ler; "er" should be italic.
 - ⇒ All instances of Ler were modified to *Ler*.

Reviewer #2 (Remarks to the Author):

The authors have addressed my comments adequately. The revised manuscript reads well.

1. I only have a minor point. I suggest authors to add the following in the discussion: At present we can not distinguish between the two possibilities whether number of WUS expressing shoot foci has increased and therefore we observe increased shoot formation or whether the expression of WUS has increased within individual foci.
 - ⇒ An equivalent statement was added to the discussion (line 335-337): *“However, at present we cannot distinguish whether improved shoot formation in Lp2-2 is due to an increased number of WUS-expressing foci or elevated WUS levels in individual foci.”*